# Intracellular *Salmonella* hijacks the mitochondrial citrate carrier to evade host oxidative defenses

Chieh-Hua Fu[1,4], Yu-Ting Hsu[1,4], Shao-Chun Hsu[2], Nai-Shu Chen[1], Hsueh-Wen Hu[1], Ting-Yin Wu[2], Yi-Jou Huang[1], Ying-Chu Chen[1], An-Chi Luo[2], Yu-Tsung Huang[3] & Shu-Jung Chang [1] ✉

Intracellular vacuolar pathogens replicate within membrane-bound compartments known as pathogen-containing vacuoles (PCVs). Maintaining the integrity of these vacuoles is essential for creating a permissive niche that supports pathogen survival and proliferation. In this study, we show that *Salmonella enterica* serovar Typhimurium co-opts the host mitochondrial citrate carrier (CIC) to promote its intracellular replication by detoxifying the *Salmonella*-containing vacuole (SCV). Loss of CIC significantly impairs *Salmonella* growth within host cells, as CIC recruitment to SCVs regulates local citrate levels and mitigates the production of reactive oxygen species (ROS), thereby reducing oxidative stress. Mechanistically, we identify the SPI-2 effector SseF as a critical factor that interacts with CIC and the GTPase RAB7, enabling CIC recruitment to the SCV membrane. These findings reveal a previously unrecognized strategy by which an intracellular pathogen hijacks a mitochondrial metabolite transporter to modulate the vacuolar environment and evade host antimicrobial defenses. Notably, pharmacological inhibition of CIC sensitizes *Salmonella* to host immune pressures, highlighting CIC as a potential target for host-directed antimicrobial therapy.

Intracellular bacterial pathogens have evolved sophisticated strategies to establish and maintain replicative niches within host cells, enabling them to evade immune defenses and exploit host resources[1,2]. A central tactic involves the formation of pathogen-containing vacuoles (PCVs), which provide a protected environment for bacterial survival and proliferation[3,4]. The establishment and maintenance of these vacuoles require precise manipulation of host organelles and intracellular trafficking pathways[4]. A key question in infection biology is how intracellular pathogens sustain vacuole integrity while avoiding degradation by host defenses.

Among these pathogens, *Salmonella enterica* serovar Typhimurium (*S.* Typhimurium) is particularly adept at intracellular survival[5,6]. As a major cause of gastroenteritis and systemic infections, *S.* Typhimurium relies on its ability to replicate within host cells, primarily inside the *Salmonella*-containing vacuole (SCV)[7,8]. The integrity and function of the SCV are essential for *Salmonella* pathogenesis, allowing the bacterium to acquire nutrients and evade cell-autonomous immunity[4,9–11]. SCV biogenesis and maintenance are driven by bacterial effector proteins delivered via type III secretion systems (T3SSs) encoded by *Salmonella* pathogenicity islands 1 and 2 (SPI-1 and SPI-2)[5,8]. These effectors remodel host trafficking pathways, ensuring vacuolar maturation, nutrient access, and avoidance of toxic insults[12,13]. SCVs have been shown to engage dynamically with host organelles, including the endoplasmic reticulum, Golgi apparatus, mitochondria, and the endolysosomal system[3,4]. Organelle-specific markers and membrane-associated proteins have been

[1]Graduate Institute of Microbiology, College of Medicine, National Taiwan University, Taipei, Taiwan. [2]Imaging Core, College of Medicine, National Taiwan University, Taipei, Taiwan. [3]Department of Laboratory Medicine, National Taiwan University Hospital, Taipei, Taiwan. [4]These authors contributed equally: Chieh-Hua Fu, Yu-Ting Hsu. ✉e-mail: sjchang@ntu.edu.tw

detected on SCVs, underscoring the intricate ways in which *Salmonella* co-opts host cell biology[14]. Yet, the functional consequences of these interactions remain incompletely understood.

In this study, we identify the host mitochondrial citrate transporter CIC (tricarboxylate carrier, encoded by *SLC25A1*) as a critical host factor recruited to the SCV during infection. We show that CIC regulates citrate export from the SCV, and that this process limits the production of reactive oxygen species (ROS), thereby detoxifying the vacuolar environment and promoting *Salmonella* replication. We further demonstrate that CIC recruitment is mediated by the SPI-2 effector SseF, which interacts with the small GTPase RAB7. Together, our findings reveal a previously unrecognized mechanism by which *Salmonella* co-opts a mitochondrial metabolite carrier to regulate the redox state of its vacuole, facilitating intracellular survival.

## Results

### CIC is recruited to the SCV in a SPI-2-dependent manner during *Salmonella* infection

A previous proteomic screen detected CIC on *Salmonella*-modified membranes[14], suggesting that CIC might be recruited to the SCV during infection. To test this idea, we transiently expressed Myc-tagged CIC in HEK293T cells, which have high transfection efficiency, and infected them for 24 h with magnetically labeled *S*. Typhimurium. SCV membranes were then isolated using a previously established magnetic-based fractionation method[15] (Fig. 1a). To ensure that magnetic labeling did not affect bacterial behavior, we assessed intracellular bacterial replication. Comparable fold-replication between magnetically labeled and non-labeled *S*. Typhimurium confirmed that magnetic labeling did not impair intracellular growth (Supplementary Fig. 1a). Additionally, expression of the SPI-2 protein PagC, monitored using a *pagC* promoter-driven superfolder GFP reporter, was indistinguishable between cells infected with magnetically labeled or non-labeled *S*. Typhimurium (Supplementary Fig. 1b). These findings validate that magnetic labeling does not alter *Salmonella* pathogenicity and supports its use for isolating functional SCVs during infection. Using this system, we found that CIC was substantially enriched in the magnetically isolated SCV fraction, where it colocalized with RAB7, a well-established SCV membrane marker (Fig. 1b). To further examine CIC localization during infection, we established a stable HeLa line expressing Myc-tagged CIC and visualized it by immunofluorescence microscopy. HeLa cells were chosen for their flat, adherent morphology, which enables high-resolution imaging of intracellular organelles. In uninfected cells, CIC-Myc colocalized almost exclusively with mitochondrial markers, confirming its expected mitochondrial distribution (Fig. 1c). Notably, we observed close spatial proximity between the SCV and mitochondria, in line with prior reports of SCV-mitochondria interactions (Fig. 1c and Supplementary Movie 1). To achieve higher spatial resolution, we employed expansion microscopy (ExM) for immunofluorescence analysis[16]. Although differential expansion of host organelle membranes and vacuolar *Salmonella* may introduce technical limitations, ExM revealed that CIC localizes in proximity to SCVs, defined as LAMP1-positive vesicles containing *Salmonella* DNA (Fig. 1d, e and Supplementary Fig. 2a–d). Importantly, this result was reproduced in cells expressing endogenous levels of Myc-tagged CIC generated via CRISPR/Cas9-mediated knock-in (Supplementary Fig. 2e, f), confirming that CIC association with SCVs is not an artifact of overexpression (Supplementary Fig. 2g, h). Notably, CIC enrichment was specific to SCVs, as we did not observe increased CIC association with LAMP1-positive late endosomes or lysosomes (Fig. 1f). Moreover, CIC in proximity to SCVs increased progressively over time post-infection (Fig. 1e and Supplementary Fig. 2i). Given that SPI-2 T3SS effectors are essential for SCV maintenance and maturation in vacuolar-residing *Salmonella*, we investigated whether they mediate CIC recruitment. Indeed, CIC accumulation at the SCV

was significantly reduced in cells infected with the SPI-2-deficient *S*. Typhimurium Δ*spiA* mutant (Fig. 1d, e and Supplementary Fig. 2g–i), indicating that SPI-2 effectors are required for this process. Collectively, these results demonstrate that CIC is specifically recruited to the SCV in a SPI-2-dependent manner during *Salmonella* infection.

### CIC is required for efficient intracellular replication of *Salmonella*

To investigate the role of CIC in the intracellular replication of *Salmonella* within host cells, we generated a CIC knockout (KO) macrophage cell line using CRISPR/Cas9 genome-editing technology (Fig. 2a). CIC deficiency did not affect cell viability (Supplementary Fig. 3a). To assess intracellular bacterial replication, we employed a fluorescence-dilution reporter plasmid (pFCcGi), previously validated for monitoring *Salmonella* replication (Fig. 2b). This plasmid encodes a constitutively expressed mCherry and a GFP under the control of an L-arabinose-inducible promoter. *S*. Typhimurium harboring pFCcGi was pre-induced with L-arabinose to express GFP and then used to infect either the parental or CIC-deficient cells. As GFP is diluted upon bacterial replication, this system allowed us to monitor intracellular proliferation by measuring GFP signal intensity over time. Our results showed that CIC inactivation led to at least a two-fold reduction in intracellular *Salmonella* replication over a 24-h period compared to parental cells (Fig. 2c). Consistent with these results, direct quantification of fold replication of *S*. Typhimurium strains SL1344 and 14028s revealed a significant reduction in bacterial replication within CIC-deficient cells (Fig. 2d). Importantly, the efficiency of bacterial entry remained comparable between CIC KO and parental cells (Supplementary Fig. 3b). Impaired intracellular growth was similarly observed across multiple cell types, including intestinal Henle-407 cells and HEK293T cells, either genetically deficient in CIC (Fig. 2e, f) or treated with a pharmacological CIC inhibitor (Fig. 2g). Notably, inhibition of CIC also significantly suppressed intracellular *Salmonella* survival in primary bone marrow-derived macrophages (BMDMs) at 24 h post-infection (Fig. 2h). Importantly, the CIC inhibitor did not affect *Salmonella* growth in broth culture (Supplementary Fig. 3c), indicating that CIC is specifically required for bacterial replication within host cells rather than for general bacterial viability.

Since CIC plays a critical role in regulating carbon metabolism and cytosolic lipogenesis by mediating citrate export from mitochondria to the cytosol[17,18], we examined mitochondrial citrate levels in CIC KO cells. As expected, the concentration of mitochondrial citrate appeared slightly elevated in CIC KO cells compared to parental cells. However, this difference was no longer observed following *Salmonella* infection (Supplementary Fig. 4a). To further assess lipid metabolism, we measured the mRNA expression of ACLY, which encodes ATP citrate lyase, a key enzyme in fatty acid biosynthesis[19,20]. We also evaluated cellular lipid content by BODIPY staining. Both analyses showed no significant differences between vehicle control and the CIC inhibitor (CTPi)-treated cells following *Salmonella* infection (Supplementary Fig. 4b, c). In addition, inhibition of lipogenesis through ACLY inactivation did not affect *Salmonella* replication (Supplementary Fig. 4d–g), indicating that the impaired *Salmonella* growth in CIC-deficient cells is not due to reduced lipid levels. We also assessed whether CIC deficiency affected cellular energy metabolism. Neither glycolytic activity nor mitochondrial respiration was significantly altered in cells treated with the CIC inhibitor (Supplementary Fig. 5). Taken together, these findings show that CIC deficiency does not appreciably alter global cellular metabolism under our experimental conditions, ruling out metabolic reprogramming as the cause of the impaired intracellular *Salmonella* growth.

Results from the cell-based models identify CIC as a host factor essential for supporting *Salmonella* proliferation within the

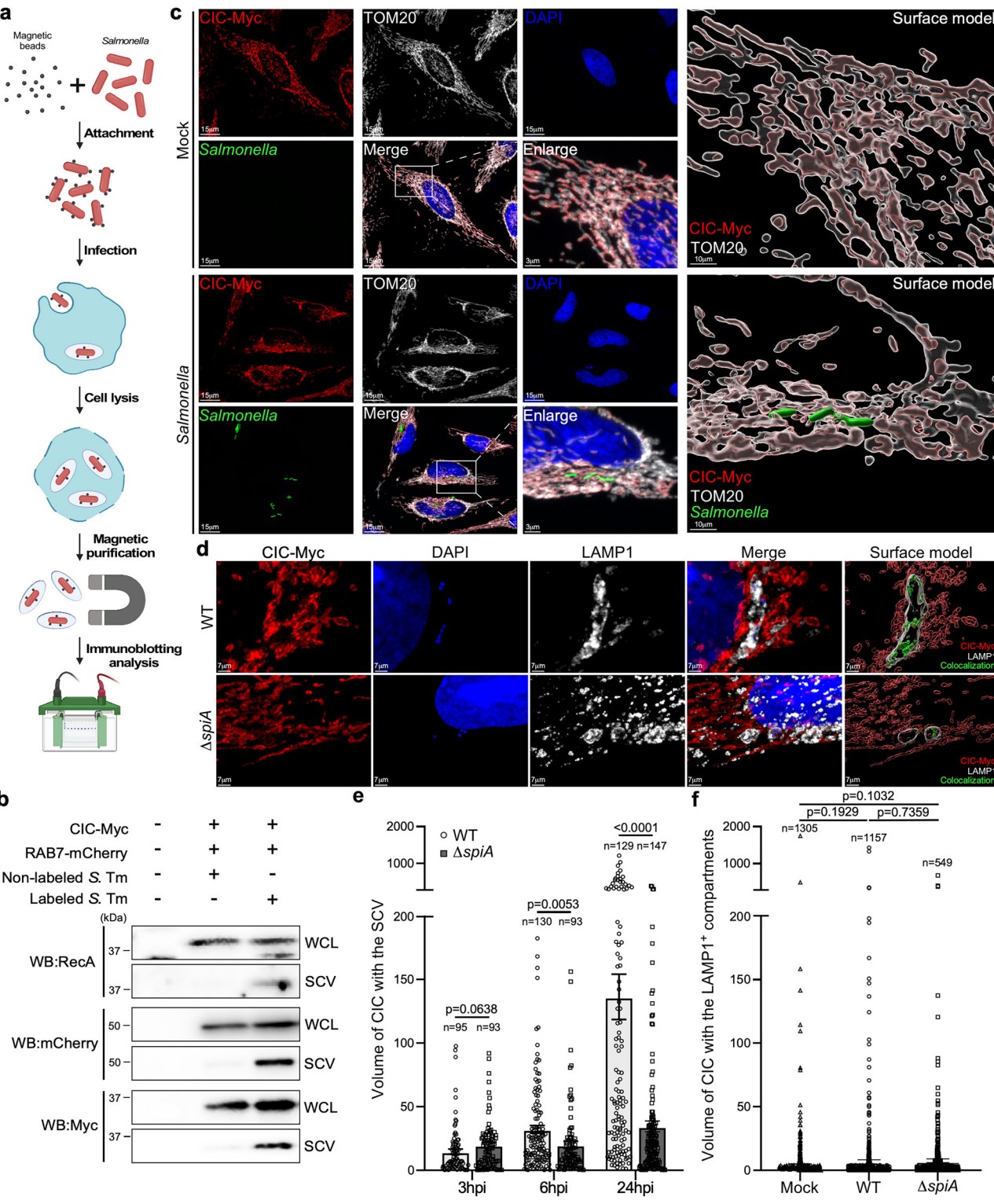

intracellular niche. To further validate its relevance in vivo, we employed a mouse infection model. Mice treated with the CIC inhibitor CTPi exhibited significantly reduced *Salmonella* burdens in both the liver and spleen (Fig. 2i). Importantly, the inhibitory effect persisted even when the compound was administered post-infection (Supplementary Fig. 6), indicating its potential as a post-exposure therapeutic intervention. Together, these findings establish CIC as a critical host determinant of intracellular *Salmonella* replication and underscore its promise as a potential target for therapeutic development.

## CIC alleviates oxidative stress for intracellular *Salmonella* within the SCV

Previous studies have shown that inhibition of CIC leads to elevated levels of mitochondrial reactive oxygen species (mtROS) in lung carcinoma cells[21–23]. However, in uninfected macrophages, mtROS levels were comparable between CIC KO and parental cells (Fig. 3a, b), and mitochondrial membrane potential remained unchanged (Supplementary Fig. 7a–d). These findings suggest that under resting conditions, CIC deficiency alone does not compromise mitochondrial membrane integrity or induce significant

**Fig. 1 | Recruitment of mitochondrial CIC to the *Salmonella*-containing vacuole. a, b** Schematic workflow for magnetic isolation of *Salmonella*-containing vacuoles (SCVs). HEK293T cells transiently expressing CIC-Myc and RAB7-mCherry were infected with magnetic bead-labeled *S.* Typhimurium SL1344 at a multiplicity of infection (MOI) of 50. At 24 h post-infection, cells were lysed and *Salmonella*-containing vacuoles (SCVs) were isolated using magnetic separation. Western blot analysis of the purified SCV fraction using antibodies against RecA (bacterial marker), mCherry (RAB7), and Myc (CIC). Cells infected with unlabeled *Salmonella* at MOI 10 served as a negative control for magnetic isolation. The experiment was independently repeated three times with representative blots shown in (**b**). The figure was created in BioRender. **c** HeLa cells stably expressing Myc-tagged CIC were either mock-treated or infected with *S.* Typhimurium constitutively expressing sfGFP. Cells were stained with DAPI, anti-Myc, and anti-TOM20 antibodies. Enlarged images show CIC localized to mitochondria in mock-treated cells and co-localizing with *Salmonella* in infected cells. Surface-rendered views further highlight CIC localization within TOM20-positive mitochondria under mock conditions

and the close spatial association of *Salmonella* (green) with CIC and mitochondria in infected cells. Three-dimensional reconstructions were generated and analyzed using the Surface module in Imaris. The experiment was independently repeated three times with a representative image show. **d–f** Post-expansion microscopy (ExM) images of cells stably expressing Myc-tagged CIC infected for 24 h with either wild-type (WT) *Salmonella* or the Δ*spiA* mutant. Cells were stained with DAPI and antibodies against Myc and LAMP1, as described in the "Methods". Three-dimensional surface models were generated using the Surface module in Imaris. Green surfaces mark regions where CIC colocalizes with SCVs. Scale bar, 7 μm (**d**). Quantification of CIC colocalization with SCVs, defined as *Salmonella* DNA within LAMP1-positive compartments, in infected cells at the indicated time points is shown in (**e**). CIC association with LAMP1-positive compartments lacking *Salmonella* is also shown for mock, WT, and Δ*spiA*-infected cells after 24 h. Data are presented as mean ± SEM of three independent experiments (**f**). Statistical analysis was performed using an unpaired two-sided *t*-test. Source data are provided as a Source Data file.

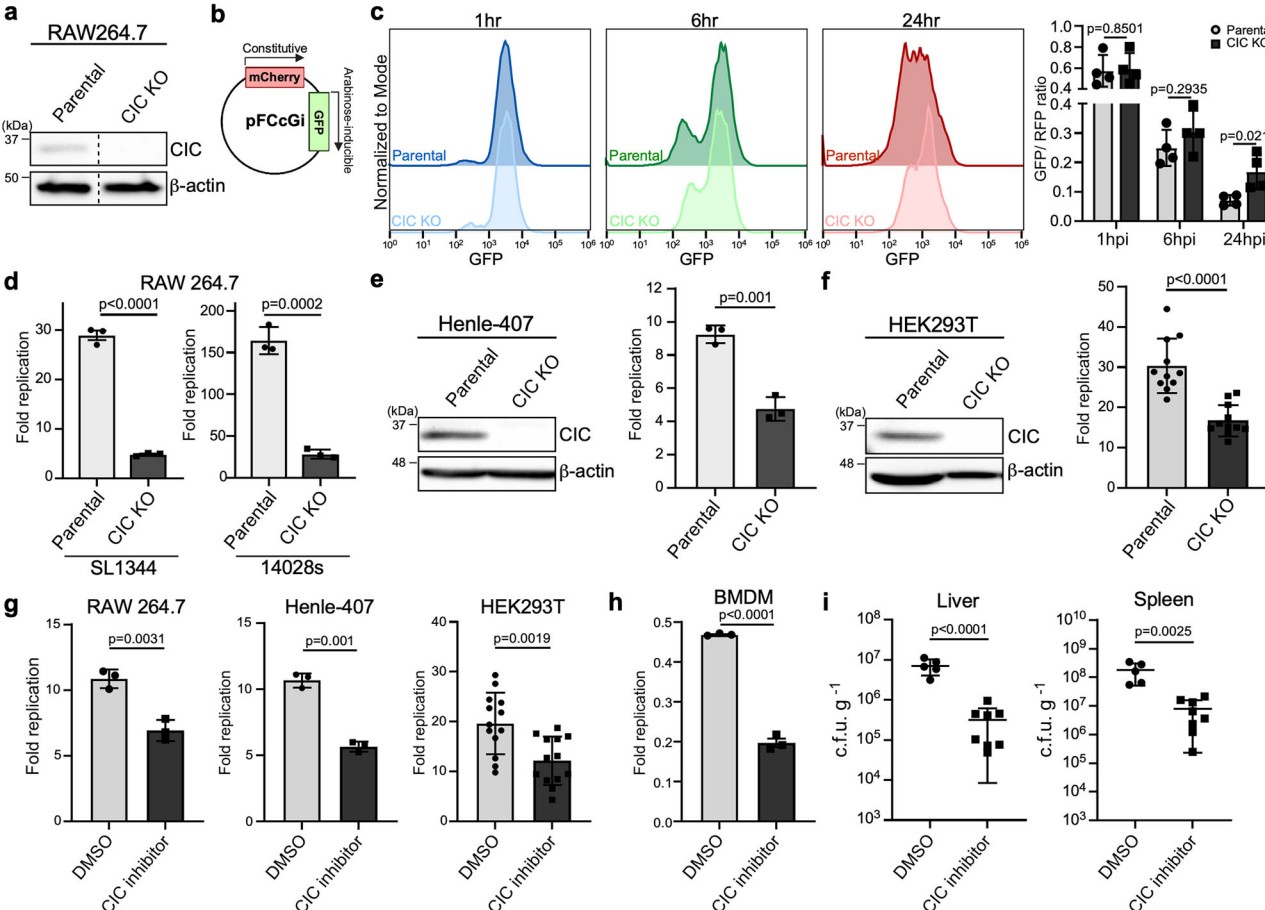

**Fig. 2 | Loss of CIC impairs the intracellular growth of *Salmonella*. a** Western blot analysis of CIC expression in parental RAW264.7 macrophages and CIC-deficient cells generated by CRISPR/Cas9 genome editing. β-actin served as a loading control. Dotted lines indicate where lanes from the same gel were spliced for presentation. **b, c** Intracellular replication of *Salmonella* in CIC-deficient macrophages assessed using the pFCcGi plasmid (**b**), which constitutively expresses mCherry and inducible GFP. Parental and CIC knockout (KO) RAW264.7 cells were infected with *S.* Typhimurium harboring pFCcGi. Bacteria were precultured with arabinose to induce GFP, then used to infect host cells in arabinose-free medium. GFP dilution relative to RFP was analyzed by flow cytometry at 1, 6, and 24 h post-infection. Replication was quantified as the GFP/RFP geometric mean fluorescence intensity ratio (**c**). Data represent mean ± s.d. from four independent experiments. **d** Intracellular growth of *S.* Typhimurium SL1344 (MOI 5) or 14028s (MOI 1) in parental and CIC KO RAW264.7 cells. CFUs were determined at 1 and 24 h post-infection, and fold replication calculated as CFUs at 24 h relative to 1 h. Data are

mean ± s.d. from three independent experiments. **e, f** CIC expression (western blot, left) and intracellular replication of *S.* Typhimurium in parental Henle-407, HEK293T, and their CIC KO derivatives. Data are mean ± s.d. from three independent experiments (**e**) and eleven independent experiments (**f**). **g, h** RAW264.7, Henle-407, HEK293T, and primary bone marrow-derived macrophages (BMDMs) pretreated with DMSO or CIC inhibitor (0.25 mM, 6 h) prior to infection (MOI 5). CFUs were determined at 1 and 24 h, and fold replication calculated as above. Data are mean ± s.d. from three independent experiments for RAW264.7, Henle-407 cells, and BMDMs, and thirteen independent experiments for HEK293T cells. **i** C57BL/6J mice were injected intraperitoneally with DMSO or CIC inhibitor and infected with *S.* Typhimurium ($10^2$ CFU). Bacterial loads in liver and spleen were measured 5 days later. Each circle or square represents CFU counts from an individual animal. Data are mean ± s.d. (unpaired two-sided *t*-test). Source data are provided as a Source Data file.

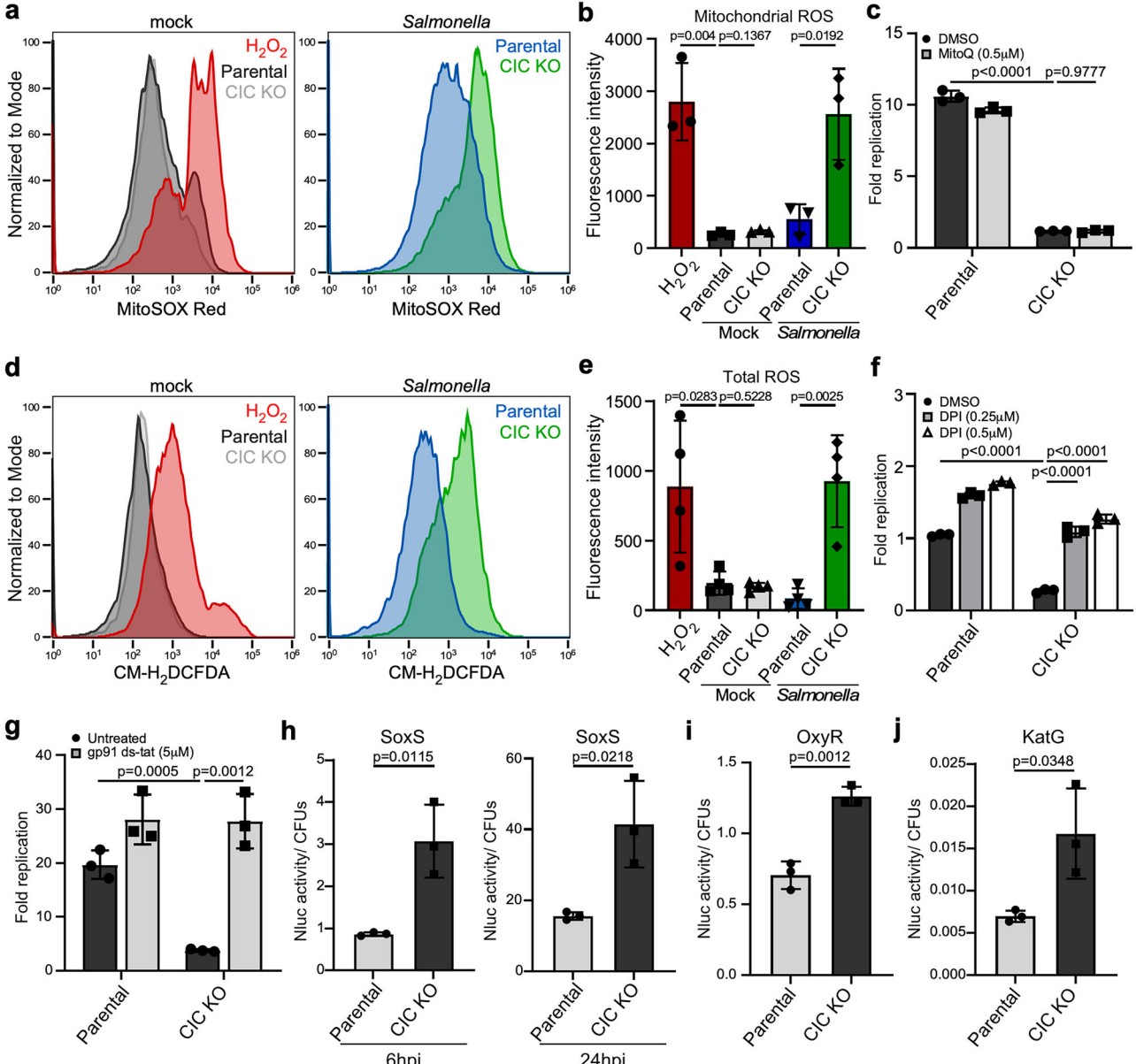

**Fig. 3 | CIC alleviates oxidative stress experienced by vacuolar *Salmonella* during host cell infection. a**, **b** RAW264.7 cells were infected with *Salmonella* Typhimurium (*S*. Typhimurium) expressing sfGFP at a multiplicity of infection (MOI) of 20. Mitochondrial ROS (mtROS) levels were detected with MitoSOX Red by flow cytometry 24 h post-infection in mock-treated and sfGFP-positive cells (**a**). H2O2 served as a positive control. Mean fluorescence intensity (MFI) values from three independent experiments are shown as mean ± s.d. (**b**). **c** Cells were infected with *S*. Typhimurium for 1 h, followed by treatment with either DMSO or MitoQ. Intracellular replication was determined as fold change in CFUs between 1 and 24 h, from three independent experiments (mean ± s.d.). **d**, **e** Total ROS levels in infected RAW264.7 cells. Cells were infected with mCherry-expressing *S*. Typhimurium at an MOI of 20. Twenty-four hours later, total ROS levels in both mock-treated and mCherry-positive cells were quantified by flow cytometry using the CM-H2DCFDA indicator. H2O2 was added as a positive control for oxidative stress. MFI values (**e**) from four independent experiments are shown (mean ± s.d.). **f**, **g** Intracellular growth of *Salmonella* in parental RAW264.7 macrophages and CIC knockout (KO) cells with or without NADPH oxidase inhibitors. Cells were infected with *S*. Typhimurium for 1 h, followed by treatment with either DMSO or DPI/ gp91phox ds-tat peptide. Fold replication represents the differences in CFUs between 1 and 24 h. Data are presented as the mean ± s.d from three independent experiments. **h**–**j** Oxidative stress in intracellular *Salmonella*. Parental and CIC KO RAW264.7 cells were infected with *S*. Typhimurium carrying nanoluciferase (Nluc) reporters driven by *soxS*, *oxyR*, or *katG* promoters. The *soxS* reporter was measured at 6 and 24 h (MOI 5), while *oxyR* and *katG* reporters were measured at 24 h (MOI 10). Luminescence represents promoter activity and is shown as mean ± s.d. of three independent experiments. Statistical analysis was performed using an unpaired two-sided *t*-test. Source data are provided as a Source Data file.

mitochondrial redox imbalance. Interestingly, upon *Salmonella* infection, we observed a marked increase in mtROS levels in CIC KO macrophages (Fig. 3a, b). Treatment with the selective mitochondrial ROS scavenger mitoquinone mesylate (MitoQ) effectively reduced mtROS levels (Supplementary Fig. 7e), but failed to rescue intracellular *Salmonella* growth in CIC-deficient cells (Fig. 3c). This indicates that increased mtROS is not the primary antimicrobial factor responsible for restricting bacterial replication in the absence of CIC.

We also measured total cellular ROS and found that while baseline levels were unchanged between CIC KO and parental cells, infection with *Salmonella* induced a significant increase in total ROS specifically in CIC-deficient cells (Fig. 3d, e). To investigate whether non-mitochondrial sources of ROS contribute to bacterial suppression,

we inhibited phagosomal NADPH oxidase (NOX) activity using either diphenyleneiodonium (DPI) or a gp91phox-specific inhibitory peptide (gp91ds-tat). Both treatments significantly restored *Salmonella* growth in CIC KO cells (Fig. 3f, g), suggesting that NOX-derived ROS are a key factor limiting bacterial replication in this context. Consistent with this, we employed a bioluminescent reporter assay in which the promoter of *soxS*, a superoxide stress-responsive gene, was fused to nanoluciferase (NanoLuc). This assay revealed that intracellular *Salmonella* within SCVs experienced elevated oxidative stress in CIC KO cells as early as 6 h post-infection, with a peak at 24 h (Fig. 3h). Using similar reporter constructs, we observed increased expression of OxyR, a hydrogen peroxide-responsive transcriptional regulator, and KatG, a key ROS-detoxifying enzyme, in CIC KO cells compared to parental controls (Fig. 3i, j). We also observed increased expression of *hmpA* mRNA (Supplementary Fig. 8), a gene involved in the nitrosative stress response. Together, these findings indicate that CIC deficiency leads to elevated phagosomal ROS, resulting in heightened oxidative stress for intracellular *Salmonella*. The consequent ROS accumulation within the SCV impairs bacterial replication, underscoring a protective role for CIC in limiting oxidative damage and promoting *Salmonella* survival within the host vacuolar niche.

## CIC modulates citrate efflux in the SCV to reduce oxidative stress in *Salmonella*

We next investigated the cause of abnormal ROS accumulation within the SCV in the absence of CIC. Mitochondrial CIC functions as a citrate carrier, exporting citrate from the mitochondrial matrix to the cytosol in exchange for malate import. If CIC performs a similar transport role at the SCV membrane, its absence may impair citrate efflux from the SCV into the cytosol, leading to local citrate accumulation (Fig. 4a). Elevated citrate levels have been associated with enhanced Fenton reaction activity[24], a spontaneous process that generates ROS, particularly hydroxyl radicals, through the interaction of ferrous iron and hydrogen peroxide[25]. Additionally, citrate accumulation can increase vacuolar acidity, further promoting ROS formation (Fig. 4a)[26,27]. To test the hypothesis that elevated ROS inhibits intracellular *Salmonella* growth as a result of increased citrate accumulation due to CIC deficiency, we first examined whether the loss of CIC leads to citrate accumulation within the SCV. In CIC KO cells, we observed significantly elevated mRNA levels of *Salmonella* genes involved in citrate sensing and metabolism (Supplementary Fig. 9a–d). These included the *cit* operon, which encodes the two-component system CitA/CitB for environmental citrate detection[28–30], as well as *citC* (citrate lyase ligase) and *citS* (citrate transporter), both of which serve as markers of citrate availability. To further validate this, we constructed a biosensor in which the promoter region of *citS* was fused to a NanoLuc reporter. Bioluminescence analysis revealed increased reporter activity in CIC KO cells compared to parental controls, supporting the notion of elevated citrate levels in the SCV of CIC-deficient cells (Fig. 4b). In addition, vacuoles harboring *S.* Typhimurium exhibited a profound decrease in luminal pH in CIC-deficient cells, as indicated by a biosensor driven by the acid stress-responsive promoter of the *stm1485* gene, which reports the acidic environment within the SCV (Fig. 4c). We also utilized a fluorescent pH-sensitive dye (pHrodo) to assess the acidity of intracellular compartments. pHrodo is a membrane-permeable probe that increases in fluorescence intensity under acidic conditions, allowing for the evaluation of vesicular pH in live cells. Fluorescence intensity in lysosomes was comparable between mock-treated parental and CIC-deficient cell lines, indicating that CIC loss does not affect lysosomal acidification (Fig. 4d, e). In contrast, vacuoles containing *S.* Typhimurium displayed markedly increased fluorescence in CIC-deficient cells (Fig. 4d, e), indicating that these SCVs exhibit a significantly more acidic environment in the absence of CIC. Taken together, these results suggest that CIC deficiency leads to elevated citrate levels and increased luminal acidity within SCVs.

We next investigated whether iron-dependent Fenton chemistry contributes to ROS production within the SCV in CIC-deficient cells. In this context, iron may be consumed through Fenton reactions, generating hydroxyl radicals and leading to local iron depletion. As a result, *Salmonella* residing in CIC-deficient SCVs could experience iron restriction and respond by upregulating iron acquisition systems to scavenge iron from the vacuolar lumen. To test this, we utilized NanoLuc-based biosensors driven by the promoters of *Salmonella iroB* and *tonB*, which are involved in siderophore biosynthesis and siderophore-iron complex uptake, respectively[31–33]. Both reporters showed significantly increased activity in CIC-deficient cells, indicating heightened expression of these iron-responsive genes (Fig. 4f, g). Consistent with these findings, RT-qPCR analysis revealed increased mRNA levels of additional siderophore-associated genes, including *entC* and *entD* (siderophore synthesis), as well as *iroC* and the *sitABCD* operon (ABC-type transporters for siderophore secretion and metal ion uptake) (Supplementary Fig. 9e–j). The induction of these genes serves as a molecular signature of iron limitation within the SCV, highlighting the adaptive response of *Salmonella* to iron-restricted conditions in CIC-deficient cells[34–36]. Notably, the *sitABCD* operon also contributes to manganese transport[37], which is critical for the activity of superoxide dismutase enzymes that protect against oxidative damage. Its upregulation may therefore represent a dual response to both iron limitation and elevated oxidative stress, aligning with the observed increase in ROS within the SCV. Importantly, supplementation with iron chelators such as deferoxamine (DFO) or polyphosphate (polyP), which sequester free iron and reduce its availability for Fenton chemistry[38], significantly enhanced *Salmonella* replication and reduced ROS accumulation in CIC-deficient cells (Fig. 4h, i). Together, these results indicate that CIC promotes vacuolar *Salmonella* growth by reducing luminal acidity and dampening iron-dependent ROS production.

## The *Salmonella* SPI-2 effector SseF orchestrates CIC recruitment to the SCV by interacting with RAB7

Given that CIC is primarily localized to the inner mitochondrial membrane, we sought to investigate the molecular mechanism underlying its recruitment to the SCV. Our observations revealed that this recruitment is dependent on SPI-2 type III secretion system (T3SS) effectors (Fig. 1d, e). Notably, a recent study using proximity-dependent biotin labeling (BioID) combined with mass spectrometry identified CIC as a potential interactor of SseF, a core SPI-2 effector known to regulate the formation of *Salmonella*-induced filaments (SIFs) and modulate host intracellular trafficking[39]. To validate the interaction between CIC and SseF, we performed co-immunoprecipitation (co-IP) assays in HEK293T cells overexpressing Myc-tagged CIC and infected them with either wild-type (WT) *Salmonella* or a strain expressing epitope-tagged SseF. The co-IP results confirmed a specific interaction between CIC and SseF in infected cells (Fig. 5a). To confirm specificity, we tested whether CIC also binds the SPI-2 effectors SopD2 or PipB2. SopD2 was of particular interest because the BioID screen suggested it as a potential interactor[39]. However, no interactions were detected between CIC and either PipB2 or SopD2 under the same experimental conditions (Supplementary Fig. 10a, b), supporting the specificity of the CIC-SseF interaction. We further validated this interaction under endogenous conditions using a CIC knock-in (KI) cell line. Co-IP analysis in this system again confirmed the association between CIC and SseF, reinforcing the physiological relevance of this interaction (Fig. 5b). To pinpoint the region of SseF required for CIC binding, we generated C-terminal deletion mutants. Prior work established that residues 195–205, which project into the host cytosol from the SCV membrane, are essential for SseF-dependent intracellular replication[40]. Consistent with this, deleting amino acids 195–205 completely abolished SseF-CIC interaction, whereas removal of the distal tail (residues 206–212) had no effect (Supplementary Fig. 10c). These results define residues 195–205 as a discrete, functionally critical

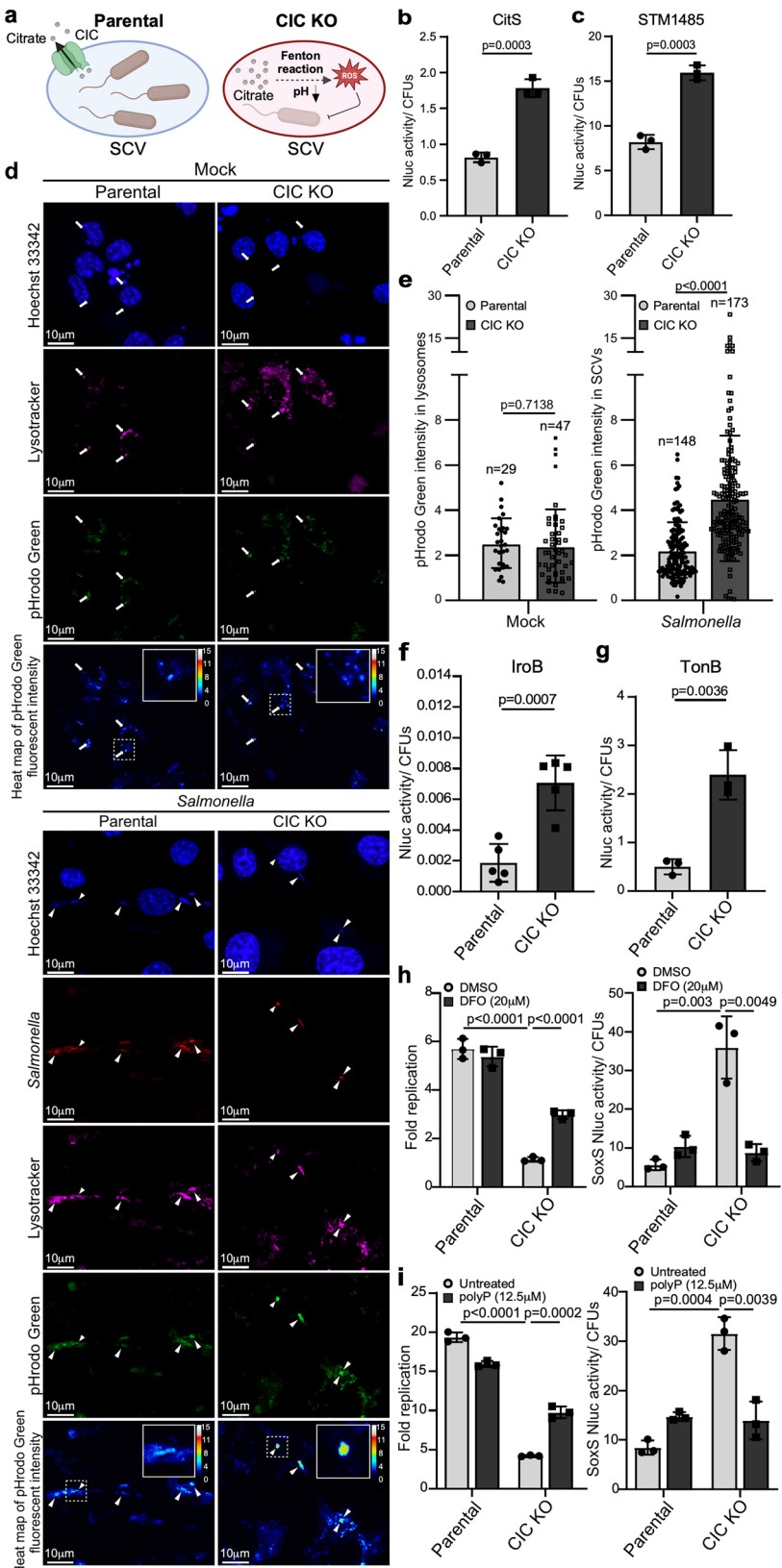

motif in SseF that mediates binding to CIC. We next sought to determine whether CIC-SseF binding is necessary for CIC recruitment to the SCV. We analyzed CIC localization in host cells expressing either Myc-tagged CIC or endogenously tagged CIC following infection with WT or Δ*sseF* *Salmonella*. In cells infected with the Δ*sseF* mutant, we observed a marked reduction in CIC localization to the SCV compared to cells infected with the WT *Salmonella* strain (Fig. 5c, d and Supplementary Fig. 10d, e). Importantly, complementation of the Δ*sseF* strain with ectopically expressed SseF-HA restored CIC recruitment to the SCV, confirming that SseF is required for this process (Fig. 5c, d). These

**Fig. 4 | CIC modulates citrate efflux in the *Salmonella*-containing vacuole (SCV) to reduce iron-dependent oxidative stress. a** Model illustrating oxidative damage caused by CIC deficiency. The figure was created in BioRender. **b, c** Nanoluciferase (Nluc) activity of *citS* and *stm1485* biosensors. Parental and CIC knockout (KO) RAW264.7 cells were infected (MOI 10) with *S.* Typhimurium carrying Nluc reporters under *citS* or *stm1485* promoters. Nluc activity was normalized to CFUs at 24 h post-infection and presented as mean ± s.d. from three independent experiments. **d, e** SCV acidity in parental and CIC KO cells. RAW264.7 cells were mock-treated or infected for 24 h with mCherry-expressing *S.* Typhimurium in the presence of pHrodo Green. Cells were stained with LysoTracker (purple) and Hoechst (blue) and imaged live. In mock cells, pHrodo fluorescence was measured within LysoTracker-positive regions (arrows); in infected cells, within mCherry- and LysoTracker-positive SCVs (arrowheads). Insets show pHrodo heat maps normalized to a 0–15 pseudo-color scale (0 = lowest, 15 = highest). Fluorescence intensities

were quantified and normalized to three randomly selected regions per image (mean ± s.d.). Data are represented as the mean ± s.d. **f, g** Nluc activity of *iroB* and *tonB* biosensos. Parental and CIC KO cells were infected (MOI 10) with *S.* Typhimurium carrying Nluc reporters under *iroB* or *tonB* promoters. Nluc activity was normalized to CFUs at 24 h post-infection. Data represent mean ± s.d. from five independent experiments (**f**) and three independent experiments (**g**). **h, i** Intracellular replication of *Salmonella* in parental and CIC KO macrophages with or without iron chelation. Cells were pre-treated with deferoxamine (DFO) (**h**) or medium-chain polyphosphate ( ~ 60 residues; polyP) (**i**) and subsequently infected at an MOI of 5. Fold replication was calculated from CFUs at 1 and 24 h from three independent experiments (mean ± s.d.). Right panels show oxidative stress levels in intracellular *Salmonella* at 24 h. Statistical analysis was performed using an unpaired two-sided *t*-test. Source data are provided as a Source Data file.

findings reveal a previously unrecognized role for SseF in facilitating the targeted recruitment of CIC to the SCV during *Salmonella* infection, likely through a direct effector-host interaction.

Intracellular vesicle trafficking is orchestrated by RAB GTPases, which are key regulators that define specific membrane identities and coordinate vesicle transport between compartments through interactions with distinct effector proteins. Recent studies have shown that the RAB7 GTPase facilitates membrane contact between mitochondria and lysosomes. Given the similarities between SCVs and lysosomes, and the well-established role of RAB7 in SCV maturation, we hypothesized that SseF facilitates the recruitment of CIC from mitochondria to the SCV through a RAB7-positive compartment. Consistent with this idea, we found that CIC forms a complex with RAB7 specifically during *Salmonella* infection (Fig. 5e), while no interaction was observed with other RAB proteins such as RAB5 or RAB34 (Supplementary Fig. 11a, b). Notably, the interaction between CIC and RAB7 was specifically induced during *Salmonella* infection and was absent in cells infected with the Δ*sseF* mutant (Fig. 5e), indicating that SseF is essential for the formation of the CIC-RAB7 complex. In support of this, we identified a multi-protein complex comprising CIC, RAB7, and SseF in infected cells (Fig. 5f). Importantly, the RAB7-SseF complex was still detected in CIC-deficient cells, indicating that CIC is not required for the interaction between SseF and RAB7 (Supplementary Fig. 11c). We also evaluated the role of the SPI-2 effector SseG, as both SseF and SseG localize to the SCV membrane and are believed to contribute to a shared virulence function during infection. However, we observed the formation of the CIC-RAB7 complex even in the absence of SseG (Supplementary Fig. 11d), suggesting that SseF alone is sufficient to mediate this interaction. Taken together, these results indicate that SseF cooperates with RAB7 in recruiting CIC to the SCV.

## Discussion

Intracellular vacuolar pathogens such as *Salmonella* exploit PCVs as replicative niches, yet host cells continuously attack these compartments by producing microbicidal ROS[41–44]. Although *Salmonella* deploys numerous SPI-1 and SPI-2 effectors to preserve vacuolar integrity and promote intracellular survival[5,45], the mechanism it uses to limit NADPH-oxidase-derived ROS has remained unclear. Here, we show that *Salmonella* recruits the mitochondrial citrate transporter CIC, normally restricted to the inner mitochondrial membrane, and relocalizes it to the SCV. CIC presence at the SCV correlates with lower citrate-dependent ROS amplification and reduced luminal acidification, consistent with a role in modulating vacuolar citrate levels. Because citrate fuels Fenton chemistry, generating highly reactive hydroxyl radicals, and also acidifies the vacuolar lumen, both effects inhibit bacterial growth[24]. Hence, CIC-mediated control of citrate provides a detoxification strategy that enhances *Salmonella* replication. Removing excess citrate therefore, appears to be an important mechanism for mitigating oxidative stress. Precise measurement of

citrate within the SCV will be an important goal for future work, yet our findings already establish CIC recruitment as a key *Salmonella* countermeasure against vacuolar oxidative stress.

CIC trafficking depends on the SPI-2 effector SseF, which interacts with the endolysosomal GTPase RAB7 during infection (Fig. 6). To visualize CIC localization, we combined biochemical enrichment of SCVs with expansion microscopy. In our protocol, host membranes expand isotropically, whereas *Salmonella* itself expands only marginally (a known limitation of current expansion microscopy for vacuolar bacteria)[46,47], producing an unavoidable size mismatch that could in principle skew apparent membrane-bacterium distances. Distortion analysis (Supplementary Fig. 2a–d) shows that host membranes retain isotropy after expansion, with a root-mean-square error of <0.1 μm over 5 μm, indicating that the SCV boundary is faithfully preserved. Consistent with the imaging, magnetically isolated SCVs are enriched for CIC together with LAMP1, confirming its vacuolar association independently of ExM. While these controls support our conclusion that CIC decorates LAMP1-positive SCV membranes, higher-resolution approaches such as correlative light-electron microscopy or protocols optimized for bacterial expansion will be valuable to define precisely how CIC associates with the SCV membrane.

Iron plays a critical role in the oxidative stress response of vacuolar *Salmonella* in the context of CIC deficiency, as it is a key element in driving the Fenton reaction[25]. Consistent with this, iron chelation markedly improved intracellular bacterial viability, suggesting that iron-dependent ROS may trigger a ferroptosis-like death pathway. Ferroptosis, originally described in tumor cells treated with erastin, is characterized by iron-driven, lipid-peroxidation-mediated ROS toxicity that occurs independently of classical apoptosis or pore-forming necrosis[48,49]. Although similar ferroptosis-like death has been reported in several microorganisms[50], the underlying mechanisms remain largely unexplored. In our experiments, however, iron chelation only partially restored *Salmonella* replication in CIC-deficient cells, implying that additional stresses persist. This incomplete rescue may reflect the indispensable role of iron in bacterial metabolism or the contribution of other ROS species generated in the increasingly acidic SCV that arises when CIC is absent. Further work will be required to dissect how iron availability, SCV acidity, and multiple ROS species together shape oxidative stress in CIC-deficient *Salmonella*.

Although mitochondrial citrate levels were not significantly elevated in CIC-deficient cells, we nevertheless explored whether other host metabolites or signaling pathways might account for the impaired intracellular growth of *Salmonella*. First, we focused on itaconate, the IRG1-derived metabolite that blocks the bacterial glyoxylate shunt[51]. If CIC loss had boosted intravacuolar itaconate, the resulting metabolic brake could have explained the replication defect. However, CRISPR/Cas9 knockout of IRG1, which eliminates

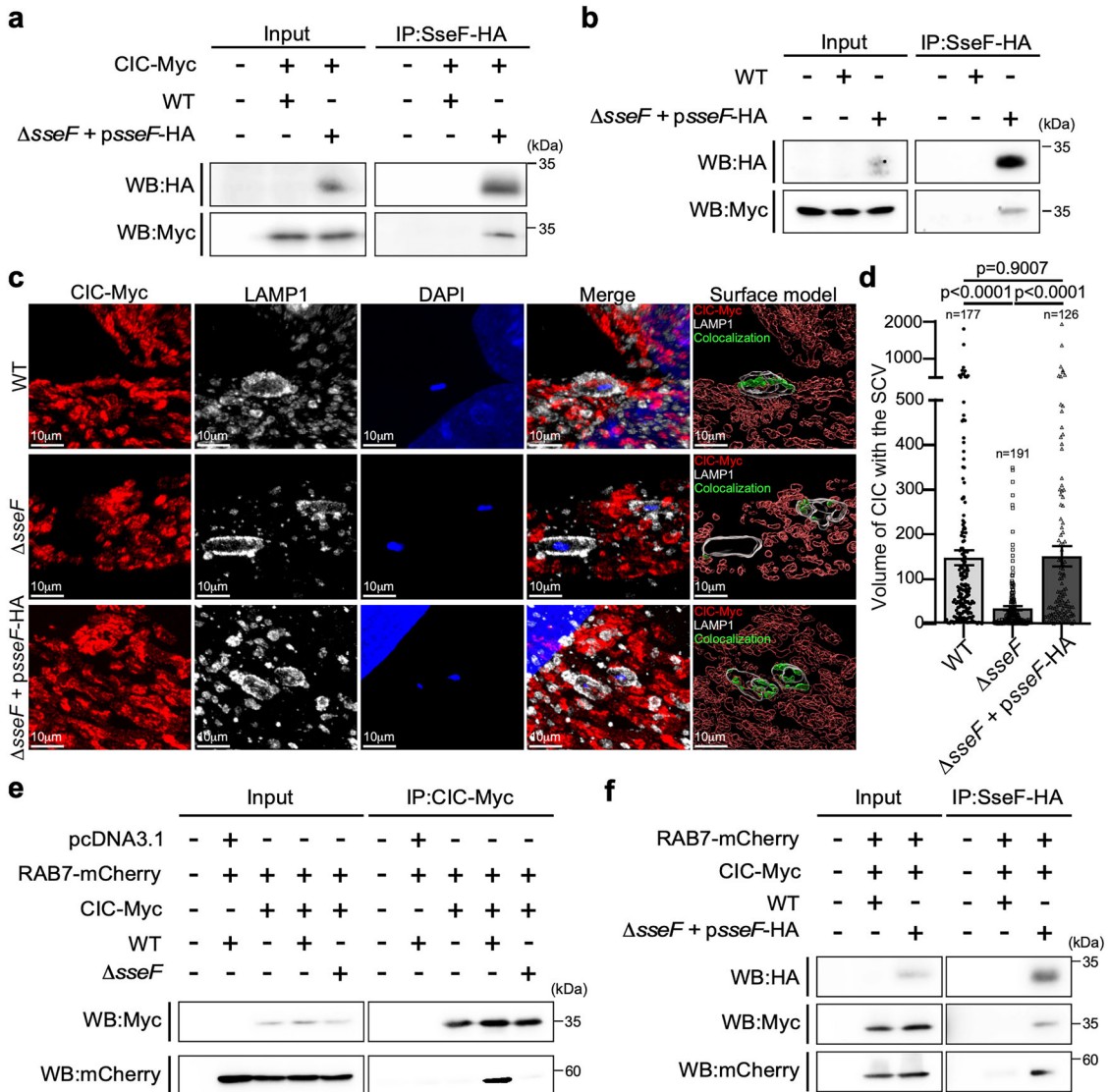

**Fig. 5 | The *Salmonella* SPI-2 effector SseF facilitates CIC recruitment to the *Salmonella*-containing vacuole through interactions with CIC and RAB7.**
**a** HEK293T cells transiently transfected with a plasmid encoding Myc-tagged CIC were infected with either wild-type (WT) *S*. Typhimurium or *S*. Typhimurium Δ*sseF* expressing HA-tagged *sseF* (Δ*sseF* + p*sseF*-HA) for 16 h (MOI = 20). Cell lysates were subsequently subjected to immunoprecipitation using anti-HA agarose, followed by immunoblotting with antibodies against HA and Myc. IP, immunoprecipitates. **b** HeLa cells with an endogenous Myc-tagged CIC knock-in were infected for 16 h (MOI = 20) with either WT *Salmonella* or a Δ*sseF* mutant expressing HA-tagged SseF. Cell lysates were immunoprecipitated with anti-HA agarose, and the precipitates were analysed by immunoblotting with anti-HA and anti-Myc antibodies. **c** Post-expansion microscopy (ExM) of CIC recruitment to SCVs. HeLa cells stably expressing Myc-tagged CIC were infected for 24 h with WT *Salmonella* (MOI = 150), a Δ*sseF* mutant (MOI = 300), or a complemented strain Δ*sseF* + p*sseF*-HA (MOI = 150). Cells were stained with DAPI (blue) and immunolabeled with anti-Myc (red)

and anti-LAMP1 (gray) antibodies, as described in "Methods". Three-dimensional surface models were generated in Imaris. Green surfaces highlight regions where CIC colocalizes with SCVs. **d** Quantification of CIC co-localization with SCVs under each condition. Data represent mean ± SEM from three independent experiments (unpaired two-tailed *t*-test). Scale bar: 10 μm. **e** HEK293T cells transiently co-transfected with plasmids encoding Myc-tagged CIC and RAB7-mCherry were infected with either WT *Salmonella* or the Δ*sseF* mutant strain for 16 h. Cell lysates were subsequently analyzed by immunoprecipitation using anti-Myc agarose, followed by immunoblotting with antibodies against mCherry and Myc.
**f** HEK293T cells transiently co-transfected with plasmids encoding Myc-tagged CIC and RAB7-mCherry were infected with either WT *Salmonella* or a complemented strain (Δ*sseF* + p*sseF*-HA). Sixteen hours post-infection, cell lysates were analyzed by immunoprecipitation with anti-HA agarose, followed by western blotting. Source data are provided as a Source Data file.

itaconate synthesis, did not rescue *Salmonella* replication in CIC KO cells (Supplementary Fig. 12). Elevated itaconate therefore, does not account for the observed growth restriction. We also examined type I/II interferon signaling, which, through STAT1, can enhance vacuolar oxidative stress by upregulating NADPH-oxidase components[52,53]. STAT1 phosphorylation was unchanged in CIC-deficient versus control cells during infection (Supplementary Fig. 13), indicating that the elevated ROS in the absence of CIC is unlikely to be driven by the IFN-STAT1 pathway.

Although CIC normally resides in the inner mitochondrial membrane, we found that it is markedly recruited to the SCV during *Salmonella* infection. This unexpected localization prompts the question of how a mitochondrial carrier reaches the vacuolar membrane. SCVs are known to tether to mitochondria and establish extensive contact sites between the two organelles[11,54], yet the molecular machinery that mediates crosstalk at these interfaces remains poorly characterized. Our data show that CIC forms a complex with the late endosomal GTPase RAB7, and that this interaction requires the SPI-2 effector SseF,

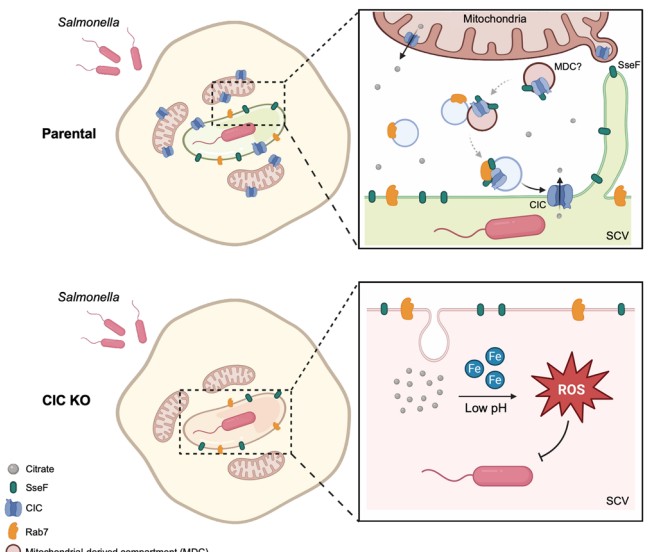

**Fig. 6 | Model of mitochondrial CIC recruitment to *Salmonella*-containing vacuoles (SCVs) and SCV detoxification.** After invasion, *Salmonella* Typhimurium remodels host endomembranes to position mitochondria in close proximity to the *Salmonella*-containing vacuole (SCV). The SPI-2 effector SseF links the late-endosomal GTPase RAB7 to the mitochondrial citrate/isocitrate carrier CIC (SLC25A1), forming a RAB7-CIC complex that promotes CIC translocation from mitochondria to the SCV membrane. Once at the SCV, CIC exports citrate/isocitrate, reducing intravacuolar reactive oxygen species (ROS). By lowering ROS, *Salmonella* evades oxidative defenses, enabling continued replication and survival within the SCV. The figure was created in BioRender.

a protein essential for vacuolar growth and SIF formation. We further demonstrate that SseF binds CIC through a short cytosol-oriented segment located at residues 195–205, a region previously implicated in effector function. These results reveal SseF as an adaptor that directs the mitochondrial carrier CIC to the SCV, thereby fortifying *Salmonella*'s intracellular niche. Furthermore, a recent study found that infection of host cells with a *sseF* mutant *Salmonella* leads to the formation of single-membrane SIFs[55,56]. This suggests that the normal structure of SIFs may be crucial for inducing the formation of the CIC-RAB7 complex, which is necessary for CIC translocation.

Moreover, RAB7 is a core component of the endolysosomal system and a common target of vacuolar pathogens [57,58]. It also mediates membrane contacts between mitochondria and lysosomes[59,60]. These observations suggest that CIC may traverse to the SCV through RAB7-positive contact regions. Analogous host-pathogen interfaces support this idea. *Toxoplasma gondii*, for example, establishes membrane contacts with mitochondria by engaging with the mitochondrial import receptor TOM70 and the outer mitochondrial membrane translocase SAM50[61]. As a result, this interaction facilitates mitochondrial remodeling, leading to the formation of structures known as SPOTs (structures positive for outer mitochondrial membrane), which support *Toxoplasma* growth. Notably, TOM70 is also required for the formation of mitochondrial-derived compartments that selectively remove SLC25A carriers under amino-acid stress in yeast[62]. These examples highlight the versatility of mitochondria-vacuole interactions and support a model in which *Salmonella* exploits an SseF-dependent pathway to relocate CIC to the SCV, thereby lowering vacuolar ROS and promoting bacterial survival.

Although SseF is the key driver of CIC translocation to the SCV, its deletion does not completely abolish CIC recruitment; a small subset of SCVs in the Δ*sseF* strain still retain CIC (Fig. 5c, d and Supplementary Fig. 10d, e). This residual recruitment suggests that

additional bacterial or host factors provide partial redundancy. Several SPI-2 effectors, including SifA, SseG, PipB2, and SopD2, are known to remodel endosomal membranes and promote the formation of double-membrane SIFs, which maintain vacuolar integrity and direct endocytic cargo to the SCV. Among these, SopD2 and SseG emerged as potential CIC interactors in the BioID screen[39]. However, our co-immunoprecipitation experiments showed that neither SopD2 nor PipB2 binds CIC, and CIC still associates with RAB7 in a Δ*sseG* background, indicating that these effectors are not required for CIC trafficking. One remaining candidate is SifA, which interacts with RAB7 and is crucial for maintaining SCV integrity[63–65]. It is therefore plausible that a SifA- and RAB7-dependent mechanism accounts for the residual CIC recruitment, perhaps by stabilizing RAB7-positive membranes or facilitating transient mitochondria-SCV contacts. Comprehensive combinatorial effector knockouts, together with high-resolution imaging of host contact-site machinery, will be needed to dissect how these parallel pathways collaborate to deliver CIC to the vacuole.

Host responses to *Salmonella* differ markedly between cell lineages. Professional phagocytes such as macrophages mount a strong respiratory burst driven by NOX and inducible nitric-oxide synthase[53,66], whereas epithelial cells rely more on pattern-recognition signaling, autophagy, and antimicrobial peptides; nonetheless, epithelial NOX can still be induced after bacterial invasion[67–70]. Across both RAW264.7 macrophages and Henle-407 intestinal epithelial cells, we show that intracellular *Salmonella* hijacks the mitochondrial citrate carrier CIC to limit ROS production within the SCV. By redirecting this transporter to the vacuolar membrane, the bacterium detoxifies its niche and enhances replication, indicating that CIC has a broad, lineage-independent role during infection.

Inoculum size also shapes host-pathogen interactions: higher multiplicities of infection (MOIs) increase bacterial entry, accelerate T3SS effector translocation, and amplify oxidative and inflammatory signals, whereas very high MOIs can provoke cytotoxicity and experimental artefacts. To minimize artefacts while meeting the technical needs of each assay, we used tiered MOIs: low (5–10) for replication and ROS analyses, intermediate (20) for flow-cytometric sorting, and high (100) solely for microscopy and intracellular *Salmonella* RNA isolation, where a large proportion of infected cells is essential for robust signal detection. Crucially, CIC recruitment to the SCV and its ROS-suppressive effect were consistent across this MOI range, confirming that our conclusions are not confined to a particular infection dose. Together, these findings reveal a previously unrecognized strategy by which *Salmonella* manipulates host mitochondrial pathways to evade antimicrobial defenses and highlight CIC trafficking as a promising target for therapeutic intervention.

## Methods
### Mouse infections
Six-to-eight-week-old C57BL/6J mice were housed in the Laboratory of Animal Center at the College of Medicine, National Taiwan University, according to protocols approved by the Institutional Animal Care and Use Committee of National Taiwan University under protocol number 20200228. Mice were maintained on a 12 h light/dark cycle (lights on from 08:00 to 20:00), at an ambient temperature of $22 \pm 2\,°C$ and relative humidity of $50 \pm 10\%$. Mice were administered $50\,mg\,kg^{-1}$ of the CIC inhibitor intraperitoneally one day before *Salmonella* infection and every other day post-infection. WT *S.* Typhimurium strain SL1344 was cultured overnight, diluted 1/20 in LB medium with 0.3 M NaCl, and grown to an $OD_{600}$ of 0.9. Mice were then intraperitoneally infected with 100 CFU of the bacteria. Five days post-infection, mice were sacrificed, and their livers and spleens were homogenized in PBS. The homogenates were serially diluted and plated on LB agar to determine CFU counts.

## Bacterial strains, plasmids, antibodies, and reagents

All *Salmonella* strains used in this study (Supplementary Table 1) are derivatives of *Salmonella enterica* serovar Typhimurium strains SL1344 or 14028s. Genetic modifications were introduced in SL1344 by allelic exchange using R6K suicide vectors with *E. coli β2163 Δnic35* serving as the donor strain for the modified alleles. All the plasmids used in this study are listed in Supplementary Table 2 and were constructed using the Gibson assembly cloning and T4 ligation cloning strategies. Antibodies, reagents, and primers used in this study are listed in Supplementary Data 1.

## Cell culture and bacterial infections

Henle-407 cells (obtained from Jorge Galan's laboratory at Yale University), HeLa cells (Bioresource Collection and Research Center in Taiwan), HEK293T cells (ATCC), and RAW264.7 cells (ATCC) were cultured in Dulbecco's modified Eagle medium (DMEM, Gibco) supplemented with 10% fetal bovine serum (FBS). BMDMs were obtained from 6- to 8-week-old C57BL/6 mice and grown in complete DMEM medium with 10% FBS and 10% L929 cell culture supernatant for 7 days. All cell lines were cultured at $37\,°C$ in a 5% $CO_2$ humidified incubator and routinely screened for mycoplasma using a standard PCR method. For *Salmonella* infection, Overnight cultures of *S.* Typhimurium strains were diluted 1/20 in LB broth medium containing 0.3 M NaCl and grown until they reached an $OD_{600}$ of 0.9. Cells were then infected with *Salmonella* strains at the MOIs indicated in the figure legends, using Hank's balanced salt solution. Infected cells were incubated with culture medium containing $100\,\mu g\,ml^{-1}$ gentamicin for 1 h to kill extracellular bacteria. After washing, the cells were incubated for the indicated times with medium containing $10\,\mu g\,ml^{-1}$ gentamicin to prevent cycles of reinfection.

## *Salmonella* growth analysis

Overnight cultures of *S.* Typhimurium were diluted 1/20 in cation-adjusted Mueller–Hinton broth and grown at $37\,°C$ until cultures reached an $OD_{600}$ of 0.9. The cultures were further diluted 1000-fold, treated with the CIC inhibitor (Cat. # SML0068, Sigma), and incubated for an additional 18 h. To determine bacterial concentrations after incubation, cultures were serially diluted 1:10, and 100 µL of each dilution was plated on LB agar plates for bacterial enumeration.

## LentiCRISPR virus production and gene ablation in cells

sgRNAs targeting genes of interest were designed, cloned into the LentiCRISPR v2-puro vector, and verified by sequencing. To generate lentivirus expressing the sgRNAs, HEK293T cells were transfected with LentiCRISPR v2 carrying the sgRNA, along with psPAX2 and pVSVg using Lipofectamine 3000 Transfection Reagent (Life Technologies, #L3000015). The culture media containing viruses were collected three days post-transfection and used for transduction on target cell lines. Transduced HEK293T and Henle-407 cells were selected with $1\,\mu g\,ml^{-1}$ puromycin, and RAW264.7 cells were selected with $5\,\mu g\,ml^{-1}$ puromycin. Two independent clones per cell line were isolated and further screened by immunoblot to confirm knockout cells. To validate the IRG1-knockout cell line, cells were stimulated with 100 ng/mL LPS for 4 h before harvest and then analyzed by western blot. These clones were characterized for the relevant phenotypes in the study, and in all cases, the different cell lines exhibited equivalent phenotypes.

## Generation of stable cell lines expressing Myc-tagged CIC

HeLa cells stably expressing Myc-tagged CIC were generated through lentiviral transduction followed by puromycin selection. Transducing viruses were produced by co-transfecting into HEK293T cells with the packaging plasmids. Three days after transfection, lentiviruses were harvested and used to transduce the HeLa cell line. Transduced cells were selected with puromycin for 2 to 3 days, and isolated clones were further tested for Myc-tagged CIC protein expression by western blot analysis.

## Generation of the Myc-tagged CIC knock-in cell line

The Myc-tagged CIC knock-in HeLa cell line was established using CRISPR/Cas9-mediated genome editing, carried out by the National RNAi Core Facility at Academia Sinica (Taipei, Taiwan). The specific sgRNA target sequences used for knock-in are provided in Supplementary Data 1.

## Luciferase reporter assay

RAW264.7 cells and Henle-407 cells were cultured in 24-well plates at densities of $1 \times 10^5$ and $7 \times 10^4$ per well, respectively. The cells were infected with different *Salmonella* strains carrying the pBAD24 plasmid encoding the nanoluciferase sensors responsive to environmental changes. An multiplicity of infection (MOI) of 5 was used for the *soxS*-driven reporter, whereas an MOI of 10 was selected for the other reporters to optimize signal-to-background ratios. Six- and sixteen-hours post-infection, cells were lysed using Passive Lysis Buffer (Promega, #N1120), and nanoluciferase levels were measured using a SpectraMax i3x Multi-Mode Microplate Reader (Molecular Devices).

## Assessment of cell death using Zombie Green staining

RAW264.7 cells were seeded at a density of $2 \times 10^5$ cells per well in 12-well plates and infected with *Salmonella* Typhimurium strain SL1344 at a multiplicity of infection (MOI) of 5. At 24 h post-infection, cells were stained with Zombie Green (BioLegend, #423111, 1:2500 dilution) for 15 min at room temperature. As a positive control, cells were fixed with 4% paraformaldehyde (PFA) for 15 min prior to staining. Samples were analyzed using an Attune NxT flow cytometer, and data were processed with FlowJo software.

## Measurement of intracellular replication rate of *Salmonella* using fluorescence dilution

*Salmonella* Typhimurium strain SL1344 encoding the dual fluorescence reporter plasmid pFCcGi (Addgene #59324) was cultured overnight at $37\,°C$ in LB medium supplemented $100\,\mu g\,\ ml^{-1}$ ampicillin and 0.2% arabinose. The bacterial cultures were spin-inoculated onto RAW264.7 cells at a multiplicity of infection (MOI) of 20, a condition that allows efficient recovery and sorting of *Salmonella*-infected cells. The fluorescent intensity of intracellular *Salmonella* within the infected cells was measured using the Attune NxT flow cytometer (Thermo Fisher), and the data were analyzed using FlowJo software.

## Lipid droplet quantification using BODIPY staining

RAW264.7 cells were seeded at a density of $2 \times 10^5$ cells per well in a 12-well plate and infected with *Salmonella* Typhimurium strain SL1344 at a multiplicity of infection (MOI) of 5. At 24 h post-infection, cells were stained with $2\,\mu M$ BODIPY (Invitrogen, #D3922) for 15 min at $37\,°C$. Samples were analyzed using an Attune NxT flow cytometer and processed with FlowJo software.

## Detection of the acidity of the *Salmonella*-containing vacuole

Lysosomal and SCV acidity was assessed using the pHrodo Green AM Intracellular pH Indicator dye (Thermo Fisher, #P35373) with slight modifications to the manufacturer's protocol. RAW264.7 macrophages were rinsed once with Live Cell Imaging Buffer (140 mM NaCl, 2.5 mM KCl, 1.8 mM $CaCl_2$, 1 mM $MgCl_2$, 20 mM HEPES, pH 7.4) and immediately infected with *Salmonella* Typhimurium strain SL1344 expressing mCherry at a multiplicity of infection (MOI) of 10 in the presence of pHrodo Green AM ($10\,\mu M$ prepared in the same buffer). Mock-treated controls were incubated with the same pHrodo Green AM–containing buffer, but without bacteria. After a 1-h incubation at $37\,°C$ in a humidified 5% $CO_2$ incubator, cells were washed with DPBS,

replenished with complete culture medium, and incubated for an additional 23 h. They were then washed once with Live Cell Imaging Buffer and counterstained in FluoroBrite DMEM (Gibco, #A1896701) containing 1 µg/mL Hoechst 33342 and 50 nM LysoTracker™ Deep Red (Thermo Fisher, #L12492). Live-cell imaging was performed using a Zeiss LSM 980 confocal microscope. Image analysis and quantification were carried out in Fiji (ImageJ). Lysosomal and SCV acidity were assessed from pHrodo fluorescence. In mock-treated cells, intensity was measured within LysoTracker-positive compartments, whereas in infected cells it was measured in SCVs positive for both mCherry and LysoTracker. All intensities were normalized to the mean fluorescence of randomly selected background regions in the same image. Raw 16-bit images (65,536 gray levels) were binned into a 0–15 pseudo-color scale (~ 4096 intensity units per bin) and displayed as heat bars, with warmer colors indicating higher acidity.

## Isolation of intracellular bacterial RNA and RT-qPCR

RAW264.7 ($1 \times 10^6$) and Henle-407 ($6 \times 10^5$) cells were seeded in 6-cm dishes and infected with *Salmonella* Typhimurium strain SL1344 at a multiplicity of infection (MOI) of 100 to maximize recovery of intracellular bacteria. To isolate bacterial RNA from infected cells, the cells were homogenized using the MagNA Lyser instrument (Roche) at 6000 rpm for 30 s and then incubated with RNAprotect Bacteria Reagent (Qiagen, #74524). Bacterial RNA was extracted using the RNeasy kit (Qiagen) according to the manufacturer's instructions, and complementary DNA was synthesized using the PrimeScript RT Reagent Kit (TaKaRa, #RR037B). Quantitative PCR was performed on the CFX96 Touch Real-Time PCR Detection System (Bio-Rad) using SyGreen Blue Mix (PCR Biosystems, #PB20.15-01). The primers used for RT-qPCR are listed in Supplementary Data 1. Data were analyzed with CFX Maestro Software (Bio-Rad) and results were normalized to the levels of the bacterial *rho* gene.

## Isolation of host cellular RNA and RT-qPCR analysis

RAW264.7 cells were seeded at a density of $2 \times 10^5$ cells per well in a 12-well plate and infected with *Salmonella* Typhimurium strain SL1344 at a multiplicity of infection (MOI) of 5. To isolate host cellular RNA, 500 µL of nucleoZOL was added to each well, and RNA extraction was performed according to the manufacturer's instructions. Complementary DNA (cDNA) was synthesized using the PrimeScript RT Reagent Kit (TaKaRa, #RR037B). Quantitative PCR was conducted on a CFX96 Touch Real-Time PCR Detection System (Bio-Rad) using SyGreen Blue Mix (PCR Biosystems, #PB20.15-01). Primers used for RT-qPCR are listed in Supplementary Data 1. Data were analyzed using CFX Maestro Software (Bio-Rad), and gene expression levels were normalized to *Actb* (β-actin) as a housekeeping gene.

## Measurement of mitochondrial citrate concentration

RAW264.7 cells were plated at $4 \times 10^6$ cells per 10-cm dish, with three biological replicates per condition. Cells were infected with *Salmonella* Typhimurium strain SL1344 at a multiplicity of infection (MOI) of 5. After 24 h, cells were harvested and mitochondria were isolated with the Mitochondria Isolation Kit (Thermo Fisher Scientific, #89874) following the manufacturer's protocol. The isolated mitochondria were resuspended in PBS and subjected to sonication (10 pulses) to release mitochondrial contents. Citrate levels were determined with the Citrate Assay Kit (Merck, #MAK333), and absorbance was recorded with a BioTek Synergy H1 plate reader.

## Seahorse extracellular flux analysis

RAW264.7 macrophages were seeded at $3 \times 10^4$ cells per well in XF24 microplates (Agilent Technologies) and infected with *Salmonella* Typhimurium SL1344 at a multiplicity of infection (MOI) of 5. The following day, oxygen-consumption rate (OCR) and extracellular acidification rate (ECAR) were measured at the specified time points using a Seahorse XF24 Extracellular Flux Analyzer (Agilent). For the Mito Stress Test, cells were washed and pre-equilibrated for 1 h at 37 °C in a humidified 5% $CO_2$ incubator with Seahorse XF DMEM assay medium (25 mM glucose, 2 mM glutamine). Compounds were sequentially injected according to the manufacturer's protocol: oligomycin (1 µM), FCCP (1 µM), and rotenone/antimycin A (0.5 µM). Basal respiration, ATP production, proton leak, and spare respiratory capacity were calculated from the OCR traces. For the Glycolysis Stress Test, cells were washed and incubated for 1 h at 37 °C in Seahorse XF DMEM assay medium supplemented with 2 mM glutamine in a non-$CO_2$ incubator. Glucose (10 mM), oligomycin (1.5 µM), and 2-deoxy-D-glucose (75 mM) were then injected sequentially. Glycolysis, glycolytic capacity, and glycolytic reserve were derived from the ECAR traces. OCR and ECAR values were normalized to cell number, and all data were analyzed using Agilent Seahorse Wave software.

## Co-Immunoprecipitation assay

HEK293T cells were seeded at a density of $2.5 \times 10^6$ cells per 10-cm dish. After 18 h, the cells were transfected with 5 µg of plasmid DNA encoding the indicated proteins or empty vectors. Eighteen hours later, they were infected with the designated *Salmonella* strains at a multiplicity of infection (MOI) of 20, a condition that maximizes detection of bacterial proteins by Western blot. Twenty-four hours post-infection, the cells were lysed in 1 ml of lysis buffer (20 mM Tris-HCl, 150 mM NaCl, 1 mM EDTA, 1% NP-40) containing protease inhibitors on ice for 30 min. The lysates were then sonicated ten times and subjected to three freeze-thaw cycles in liquid nitrogen to enhance cell lysis. Following centrifugation at $16,000 \times g$ for 10 min at 4 °C, the supernatants were incubated with 20 µl of Sepharose agarose (50% slurry; MERCK #GE17-0780-01) for 1 h at 4 °C to reduce non-specific binding. The pre-cleared supernatants were then incubated with 20 µl of prewashed anti-HA agarose (50% slurry; Sigma #SI-A2095) or anti-Myc agarose (50% slurry; Croyez Bioscience #C07001) for 3 h and 16 h, respectively, at 4 °C. The immune complexes were collected by centrifugation at $5000 \times g$ for 30 s, washed four times with lysis buffer, and resuspended in 40 µl of Laemmli sample buffer. The samples were analyzed by SDS-PAGE and western blot using antibodies as indicated in the figure legends.

## Immunofluorescence staining

HeLa cells ($3.5 \times 10^4$ per well) were infected with *Salmonella* Typhimurium SL1344 carrying a superfolder-GFP plasmid at a multiplicity of infection (MOI) of 100. This higher MOI was chosen to offset the relatively low invasion efficiency of *Salmonella* in HeLa cells and to yield sufficient SCVs for quantitative analysis. Twenty minutes after infection, cells were fixed in 4% PFA for 20 min at room temperature. After three washes with PBS, cells were permeabilized with 0.1% saponin in PBS for 1 h at room temperature and incubated overnight at 4 °C with primary antibodies against Myc (ABclonal; 1:1000) and TOM20 (Santa Cruz Biotechnology; 1:500). Following additional washes, cells were stained for 1 h at room temperature with Alexa Fluor 594- and Alexa Fluor 647-conjugated secondary antibodies (Invitrogen) and counterstained with DAPI. Images were acquired using a Zeiss LSM 780 confocal microscope.

## Isolation of *Salmonella*-containing vacuoles

This assay was adapted from Singh et al. with minor modifications[15]. Briefly, HEK293T cells were seeded at $1.5 \times 10^6$ cells per 6-cm dish and grown for 18 h before transfection with 2 µg of the indicated plasmid DNA. Carboxyl-coated paramagnetic cobalt magnetic beads (10–50 nm; TurboBeads, Zurich) were suspended at 3 mg mL$^{-1}$, dispersed by two 2-s sonication pulses, centrifuged at $1000 \times g$ for 90 s, and the supernatant was diluted 1:10 with distilled water. A 1-mL pellet (~ $1 \times 10^9$ CFU) of *Salmonella* Typhimurium SL1344 was resuspended in DPBS, mixed with magnetic beads at a 1:5 (bacteria:bead) ratio, and

incubated for 90 min at 37 °C to generate magnetically labeled bacteria. Two and a half hours post-transfection, cells were infected with labeled or unlabeled bacteria at the multiplicity of infection (MOI) specified in the figure legends. Twenty-four hours later, cells were lysed on ice in 1 mL of lysis buffer (50 mM Tris-HCl, 150 mM NaCl, 0.1% Triton ×-100, protease inhibitors) for 30 min, sheared by five passes through a 27-gauge needle, and diluted 1:1 with the same buffer. Lysates were subjected to a magnetic field for 2 min to isolate bead-bound *Salmonella*-containing vacuoles, washed once with DPBS, resuspended in RIPA buffer, and analyzed by SDS-PAGE and immunoblotting with the antibodies listed in the figure legends.

## Expansion microscopy

HeLa cells stably expressing Myc-tagged CIC were infected with *Salmonella* Typhimurium. At the time points indicated in the figure legends, cells were fixed in 4% PFA for 20 min at room temperature and immunostained with primary antibodies against Myc (ABclonal, 1:1000) and LAMP1 (Cell Signaling, 1:100). After washing, cells were incubated with anti-biotin secondary antibody (Thermo Fisher, 1:200) and STAR 580 (1:200), followed by Alexa Fluor 488 Streptavidin (Thermo Fisher, 1: 200). After staining, cells were post-fixed with 0.1% glutaraldehyde for 1 h at room temperature and incubated overnight with 100 µg/mL acryloyl-X, SE (6-((acryloyl)amino)hexanoic acid, succinimidyl ester; AcX) at room temperature. On the following day, cells were incubated on ice for 5 min with gelation solution (8.6% sodium acrylate, 2.5% acrylamide, 0.15% N,N'-methylenebisacrylamide, 2 M NaCl, 0.1% TEMED, and 0.1% APS in PBS), then transferred to a gelation chamber and incubated at 37 °C for 1 h to allow polymerization. The samples were subsequently incubated overnight at room temperature in proteinase K digestion buffer (0.5% Triton ×-100, 1 mM EDTA, 50 mM Tris, 1 M NaCl, 8 U/mL proteinase K) to degrade structural proteins. On the final day, the gels were expanded by incubating in water for 10 min, repeated three times. The expanded gels were then stained with DAPI (1.3 µg/mL in water) for 30 min at room temperature and washed twice with water for 10 min each. For expansion microscopy of HeLa cells expressing CIC-Myc at the endogenous locus, the same fixation and primary-antibody steps were followed. Samples were then incubated with anti-biotin (1:200) and Alexa Fluor 568 Streptavidin (1:200), followed the next day by a second round of anti-biotin and Alexa Fluor 488 Streptavidin (1:200) for sequential labeling. After staining, gels were processed with the same expansion protocol described above. Images were acquired using a Leica TCS SP8 X STED 3× microscope and processed with Imaris software (Oxford Instruments) for analysis. The tool used to analyze SCVs in ExM images is available at: https://github.com/peggyscshu/SCV-extractor.

## Detection of cytosolic and mitochondrial reactive oxygen species

Cells were infected with *Salmonella* Typhimurium expressing either mCherry or superfolder GFP (sfGFP) at a multiplicity of infection (MOI) of 20, a dose that permits efficient recovery and sorting of infected cells. Cellular and mitochondrial ROS levels were measured using 2′,7′-dichlorofluorescein diacetate (DCFDA) (Thermo Fisher, #C6827) and MitoSOX (Invitrogen, #M36008), respectively, following the manufacturers' instructions. The samples were analyzed using an Attune NxT flow cytometer and FlowJo software.

## Software

The following software was used in this study: CFX Manager™ (version 4.1.2433.1219) for gene expression analysis; ChemiDoc MP Imaging System (Bio-Rad) together with Image Lab (version 6.0.1) for western blot imaging and band quantification; Imaris (version 9.9.1, Oxford Instruments) and ImageJ (version 1.54f) for image analysis; Canvas X Draw (version 20) and Adobe Photoshop (version 26.6.0) for image preparation; FlowJo (version 10.9.0) for flow cytometry analysis;

Agilent Seahorse Wave Desktop (version 2.6.3.5) for Seahorse XF data analysis; Agilent BioTek Gen5 (version 3.16) for citrate measurement; SoftMax Pro (version 7.1.0) for NanoLuciferase assays; and BioRender for creating scientific illustrations.

## Statistics and reproducibility

All experiments were independently repeated at least three times, with reproducible and consistent results. Statistical analyses were performed using GraphPad Prism 9. Student's two-tailed *t*-test was used to assess statistical significance. Exact *P* values are reported and indicated in the figures. Error bars represent either the standard deviation (SD) or the standard error of the mean (SEM), as specified in the figure legends. Sample sizes were not predetermined by statistical methods. No data were excluded from the analyses.

## Reporting summary

Further information on research design is available in the Nature Portfolio Reporting Summary linked to this article.

# Data availability

Source data are provided with this paper.

# Code availability

The codes used in this study are available on GitHub and archived on Zenodo (DOIs: https://doi.org/10.5281/zenodo.13373734[71]. https://doi.org/10.5281/zenodo.13756849[72]).

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

## Acknowledgements

We sincerely thank Dr. Toshikazu Shiba (RegeneTiss, Japan) for the generous gift of EXCLUSIVE POLYPHOSPHATE (medium-chain). We thank the staff of the Biomedical Resource Core at the First Core Labs, College of Medicine, National Taiwan University, for technical assistance. We thank Dr. Yu-Chi Chou and the RNA Technology Platform and Gene Manipulation Core Facility (RNAi core) of the National Core Facility for Biopharmaceuticals at Academia Sinica in Taiwan for providing CRISPR related services. We also thank the staff of the Seventh Core Lab, Department of Medical Research, National Taiwan University Hospital for technical support during the study. This work was supported by the Taiwan National Science and Technology Council grants NSTC113-2636-B-002-004 (S.J.C.) and NSTC113-2320-B-002-076 (S.C.H.), the Yushan Fellow Program by the Ministry of Education (MOE) (NTU113V1404-5 [S.J.C.] and NTU-114V2014-1 [S.J.C.]), and NTU-114L7858 (S.J.C.). Schematic illustration created with BioRender.

## Author contributions

S.J.C. conceived and directed the project. S.J.C., C.H.F., and Y.-T.Hs. designed the experiments and interpreted the data. C.H.F. and Y.-T.Hs. performed most of the experiments. S.C.H., N.S.C., H.W.H., Y.-J.H., and Y.-C.C. conducted additional experiments and contributed to data interpretation. S.C.H., T.Y.W., and A.C.L. contributed to the expansion microscopy experiments and image analysis. Y.-T.Hu. supervised experiments and assisted with data interpretation. S.J.C. wrote the manuscript with input from all authors.

## Competing interests

The authors declare no competing interests.
