## [Transparent Peer Review file · Nature Communications]

Intracellular *Salmonella* hijacks the mitochondrial citrate carrier to evade host oxidative defenses

Corresponding Author: Professor Shu-Jung Chang

Version 0:

Reviewer comments:

Reviewer #1

(Remarks to the Author)

In this study, the authors showed that the mitochondrial tricarboxylate carrier protein (CIC) is recruited to the SCV in *Salmonella*-infected cells. They proposed CIC to play a role in limiting ROS production by the host, promoting *Salmonella* intracellular survival and growth. Finally, they identified the SseF SPI-2 effector to be involved in the recruitment of CIC to the SCV.

The results are very interesting and particularly important for understanding the intracellular mechanisms by which *Salmonella* adapts to its vacuolar environment. The proposed model is essentially based on microscopy, infection, regulation and localisation experiments aimed at demonstrating that the presence of CIC on the SCV significantly reduces ROS production by the host and limits its antibacterial activity against intracellular *Salmonella*.

There is no doubt about the quality of the experiments and their novelty. However, I have a few comments, most of which focus on Figure 3, which presents results that are central to arguing and validating the main message and title of this article. From my point of view, a certain confusion and uncertainty persists after reading this section.

1. The wording, the legends and the organisation of the figures are rather confusing and unintuitive. Overall, it would be good to describe the results more explicitly to help the reader follow the experimental progress. In addition, the names of the probes studied or the reporters used are specified in the legends but not on the figure, which complicates and slows down understanding of the figure. Finally, the panels are not always numbered in the right order.

2. ROS are measured using the CM-H₂DCFDA indicator, which does not allow the type(s) of ROS measured to be distinguished. It would therefore be appropriate to use other probes or biosensors specific to one form of ROS and to specify which one, even if a soxS reporter is used.

3. The qRT-PCR measurements carried out on the katG gene show a relatively modest increase (about 1.2-fold 6h pi and 2.8-fold 24h pi) which is not in agreement with the differences in ROS levels observed in figures 3b (about 20-fold) and 3d (about 8-fold). Do the authors have an explanation?

4. DPI is an inhibitor of flavoenzymes, including NADPH oxidase. But it would probably be more convincing to carry out these same experiments in gp91^{-/-} macrophages lacking the catalytic subunit of this complex and therefore unable to produce superoxide anion or any other ROS.

Minor points

Lines 113-115: This result is not supported by Figure S3 where the addition of CIC was not significant in the liver and spleen.

Lines 192-193: This is highly speculative and should be mitigated: all the experiments used to support this conclusion are indirect.

Reviewer #2

(Remarks to the Author)

Comments to the authors:

The manuscript Fu et al. proposes a novel mechanism of recruitment of host cell mitochondrial citrate transporter CIC to the Salmonella-containing vacuole (SCV) in Salmonella-infected mammalian cells. The work shows association of CIC with SCV membrane, and the authors propose functional insertion of CIC in the SCV membrane to reduce citrate concentrations in the SCV lumen. Reduction of citrate in the SCV in turn should limit Fenton chemistry reactions and protect intracellular Salmonella from oxidative damages. SPI-2 effector protein SseF is proposed to mediate the recruitment of CIC for inner mitochondrial membrane to the SCV membrane.

This is an interested novel observation, but a large number of experimental limitations and biased interpretations limits my enthusiasm about this work.

Specific comments:

In Fig. 1, no description of the magnetic conjugation and separation protocol is provided. According to the legend, Salmonella were conjugated with magnetic beads, then used in invasion of HEK293T cells, followed by magnetic isolation of Salmonella containing compartments. Although the description of materials and methods is rather comprehensive for other parts of the manuscript, this procedure is not described at all. It appears technically rather challenging to conjugate magnetic beads to Salmonella in a way that is compatible to cell invasion. Please provide sufficient details for the procedure and show controls.

The visualization and quantification of CIC association with SCV (Fig. 1cde) requires controls. The association of LAMP1-positive compartments (late endosomes/lysosomes) with CIC-positive compartments needs to be determined under comparable conditions for non-infected cells and cells infected with SPI-2-deficient Salmonella.

Does the protocol used for expansion microscopy also allow expansion of the bacterial cells? In Fig. 1 the labeled host cell compartments appear expanded, but not the bacteria within the SCV. If this is the case, the interpretation of potential contact between SCV membrane positive for LAMP1 and mitochondrial membranes positive for CIC has to be taken with care. To make valid conclusions about a potential colocalization of CIC with CIC membranes, or even the integration of CIC into SCV membranes, more detailed analyses are required.

A Salmonella strain set was used that suffers from atypical cell division (septation defect). Note the appearance of filamentous Salmonella cells in Fig. 1 and Fig. S9. This phenomenon has been observed previously and was explained by the His auxotrophy of SL1344. Specific care should be taken in the interpretation of CFU assays for intracellular survival and replication, as filament formation will result in incorrect CFU counts.

Fig. 2: What is the effect of CIC k/o and CIC inhibitor on physiology of host cells? Give that the mitochondrial transporters are central for mammalian cell physiological, critical assessment is required. Is host cell viability altered? Is phagocytic uptake or invasion by Salmonella affected? How is fatty acid metabolism altered in k/o cells or cell exposed to the inhibitor?

Fig. 4 fg: pHrodo-labelled Salmonella were used to monitor the SCV pH in dependency of CIC function. This is basically a useful experimental approach. Criticism here are in particular: a) if bacteria are prelabelled with a pH sensor, intracellular replication of Salmonella will gradually dilute the label. This issue may be limited at 6 h after infection (p.i.), but more prominent at 24 h (experiments shown in Fig. 1 demonstrate the dilution during replication in RAW macrophages). Thus, in absence of normalization, the pHrodo cannot be interpreted. b) As I understand from the description of this experiment, no normalization for loading of RAW cells with Salmonella was performed. Again, interpretation of the signal intensities is hardly possible. If the aim here is to correlate the CIC function to phagosomal pH, I suggest to use pHrodo labelled beads (that does not replicate within host cells).

X and Y scaling are missing in Fig. 4f (as for most flow cytometry histograms).

Fig. 4h-o: Here the authors analyse the expression of iron-regulated genes in intracellular Salmonella, and the data indicate iron limitation for Salmonella in CIC k/o cells. As other analyses were performed with Salmonella as biosensors, it would be of interest to use a biosensor approach also for analyses of iron limitation of Salmonella within the SCV.

Fig. 4: Is there any evidence for increased Fenton chemistry in CIC k/o cells? If this assumption is valid, oxidative damages should be more frequent in CIC k/o cells independent from Salmonella infection.

The model also proposes that citrate export from the SCV by CIC is critical for the protection of Salmonella against oxidative damage. To appreciate the important of citrate concentration in the SCV, it would be important to determine the citrate concentration in the SCV in WT and CIC k/o cells, as in response to intracellular activities of Salmonella. Without such direct data, the models on the role of citrate in the SCV are rather speculative.

Fig. 5: The data shown in Fig.5 should demonstrate that SPI-2 effector SseF is mediating CIC recruitment to the SCV. This assumption of mainly based on a previous BioID approach, but not by systematic analyses of contribution of the various SPI-2 effectors.

Complementation of the phenotype of the sseF mutant strain shown in Fig. 5b is required.

Experiment shown in Fig. S8 should demonstrate that SPI-2 effector proteins SopD2 and PipB2 are not involved in CIC recruitment to SCV. However, the IP results are difficult to compare to IP data shown in Fig. 5. For SseF, the HA tag was used for IP, and Fig. 5 indicates that SseF-HA is efficiently translocated into host cells and recovered by IP. This does not apply to SopD2-FLAG and PipB2-FLAG, as Figure S8 indicated very low levels of translocated effects. The experiment needs to be repeat with HA-tagged effectors to allow comparison and proper interpretation.

All conclusions on CIC localization are based on transiently or permanently transfected host cells expressing tagged allele of CIC. It will be important to corroborate the findings made in the overexpression system with analyses of CIC expressed at endogenous levels, either using antibodies against CIC or an endogenously tagged allele of CIC.

Fig. 6 and overall model: CIC is located in the inner mitochondrial membrane, and the model developed by the authors propose that CIC is recruited to the SCV membrane. Such mechanism would require extraction of CIC from the mitochondrial inner membrane, transfer to the SCV membrane, and membrane insertion. The scheme in Fig. 6 proposes formation of vesicles that also contain Rab7 and SseF. How can such compartment be formed?

Reviewer #3

(Remarks to the Author)

In the article entitled 'Intracellular Salmonella hijacks the mitochondrial citrate carrier to evade host oxidative defenses', Fu et al present a compelling study investigating the role of the mitochondrial citrate carrier (CIC) in the intracellular survival and replication of Salmonella Typhimurium. The authors demonstrate that CIC is recruited to the Salmonella-containing vacuoles (SCVs), where it plays a crucial role in mitigating oxidative stress. This recruitment is mediated by the Salmonella effector protein SseF and the host GTPase Rab7. The authors state CIC alleviates oxidative damage by modulating citrate levels within the SCVs, thereby promoting bacterial survival and replication. Pharmacological inhibition of CIC significantly reduces bacterial replication and colonisation, suggesting CIC as a potential therapeutic target.

While the study provides valuable insights into host-pathogen interaction eventually introducing novel host-directed strategies for Salmonella, my enthusiasm for the manuscript would greatly increase if several critical shortcomings were addressed. For example, alternative hypotheses to CIC deletion impact on host metabolism and related antimicrobial response are not explored. The data generally support the claims, but additional evidence is needed for certain mechanistic aspects (e.g., citrate levels, SseF domains).

Major:

1. A main concern is that the description of some experiments lacks sufficient detail, making it challenging for the reader to follow. For example, the explanation of the fluorescence dilution experiments in Figure 2 is overly vague. There is no clarification of the two colours used, which one is inducible, what the fluorescent dilution signifies, or what the graphs show. Additionally, it is somewhat unclear how fold replication was calculated in this figure and whether this method is validated when using this tool. The figure 1 microscopy images don't look compelling, perhaps a better description would help

2. Although the authors link citrate regulation to ROS mitigation, there is no direct measurement of citrate levels in SCVs or their correlation with ROS production. Similarly, while iron depletion is suggested as a key factor, the authors provide only indirect evidence.

The authors should isolate SCVs and employ metabolomic or targeted biochemical assays to measure citrate and iron concentrations. Additionally, to confirm the role of the Fenton reaction, direct inhibition using polyphosphate (a potent inhibitor in vitro) should be tested (10.1128/mbio.01017-20).

3. How does CIC inhibition/depletion, used in several experiments, affect citrate concentration in mitochondria? Citrate is involved in TCA cycle, its accumulation in CIC KO cells could lead to enhanced oxidative phosphorylation. Cell metabolism pathways such as glycolysis and OXPHOS (and mtROS) have been repeatedly linked to interferon response mounting (10.1016/j.cell.2019.05.003, 10.1126/scisignal.aaf1933, 10.3390/v16091391) which is known to inhibit Salmonella replication. It is then possible to argue that the inhibition observed in CIC KO cells is in part linked to a such effect. In a similar manner, citrate and TCA cycle are involved in the production of itaconate, a potent antimicrobial metabolite. It is possible to argue that accumulation of citrate in the mitochondria of CIC KO cells could lead to the a change in itaconate production (10.1016/j.celrep.2022.110391) therefore limiting Salmonella replication.

An alternative (or complementary) mechanism to the one presented in this work is highly possible and should be explored (or at least discussed) given that the level of Salmonella growth did not reach the level of parental cells after inhibition of cell ROS in CIC KO cells.

4. The authors strongly assert that SseF is the sole mediator of CIC recruitment. However, data from Figure 5B and Supplementary Figure S9 show that CIC recruitment persists, albeit at reduced levels, in the absence of SseF, indicating additional mechanisms. Revise the assertion regarding SseF's exclusivity in CIC recruitment. The role of other SPI-2 effectors or host factors should be discussed.

Also, the mechanistic details by which SseF and Rab7 facilitate CIC recruitment remain elusive. Mutational analyses of SseF to identify critical domains necessary for interaction, could be done or at least discussed.

5. One of my major concerns is the inconsistency in the methods and cells used. It raises concerns about reproducibility and interpretation of the data. At several time in their work the authors changed the MOI used (MOI 5 in Figure 2, MOI 20 in Figure 3, MOI 100 in Figure 4...) for infection as well as the cells used for their assays, without justifying it or discussing it. These changes have fundamental consequences since the MOI used could affect the response elicited by the host. Furthermore, the infection in epithelial cells (HeLa) and non-epithelial cells (RAW264.7 and BMDM) is fundamentally different. An explicit rationale for the use of different MOIs and cell lines in each experiment should be provided. Moreover, the authors should discuss how these differences might influence host responses and Salmonella replication. Also, ensure that all experiments specify the cell type used. This would make it easier to interpret the results, which are currently confused.

Minor:

1. Figure 1C: The authors stated that they noted the close proximity between SCV and mitochondria, as well as CIC localization on mitochondria in non-infected cells. However, no mitochondrial marker was used in these experiments and images with mitochondrial marker should be provided.

2. Figure 1E: Quantitative data are presented, but the recruitment dynamics could be more clearly visualized. Add an illustrative image or graph summarizing CIC recruitment over time.

3. Figure 2G: The reduction in bacterial replication may result from cell death rather than inhibition of Salmonella. An assessment of the cytotoxicity of the CIC inhibitor at the concentrations used in the study, should be provided.
4. Line 112: "in vivo" should be in italics.
5. Figure 3: ROS is highly labile, making static measurements potentially misleading. Is it possible to consider live imaging for ROS measurements? Live imaging could therefore permit to have a better understanding of ROS dynamics in the different conditions tested.
6. Line 205: "in vivo" should be replaced by "in vitro".

Reviewer #4

(Remarks to the Author)

Version 1:

Reviewer comments:

Reviewer #1

(Remarks to the Author)

I have read the new results and the changes made by the authors. My suggestions have all been taken into account and my concerns resolved after a lot of experimental work and rewriting of many paragraphs.

I would like to congratulate the authors on their work which, in my opinion, makes their publication much sounder and a much more flowing read.

Reviewer #2

(Remarks to the Author)

Thank you for the careful revision and detailed response to my comments. The authors have devised and executed a large number of additional experiments in order address the comments made to the original version of the manuscript. The new data added to the manuscript significantly improve the overall quality of the study and the robustness of interpretation. For this reviewer, two points previously made still require attention:

1) Expansion microscopy for colocalization analyses: A correctly responded by the authors, the degree of expansion of host cell organelles and intracellular Salmonella is different. The method is use to address, with increased resolution, proximity between host cell membrane compartments. LAMP1-positive compartment can be Salmonella-containing vesicles, or host cell compartments that are not modified by activity of intracellular Salmonella. Thus, conclusions based on colocalization between LAMP1 and mitochondrial marker remain of limited value. I suggest to adjust the interpretation of this experiment and to tone down the statements based to expansion microscopy data. Alternatively, if the authors wish to build on expansion microscopy, markers other than LAMP1 would be required, for example SPI-2 effector proteins such as SseF that specifically insert in SCV and SIF membranes. As effector proteins as markers will only work in infected cells, comparison has to be made between wild-type and mutant strains.

2) Recruitment of CIC to SCV: in the initial set of comments, I asked how recruitment of CIC from inner mitochondrial membranes to the SCV membrane can be mechanistically explained. The authors provide a set of potential routes such s formation of mitochondria-derived vesicles as potential vehicle. Although currently speculative and beyond the range of investigations in this current manuscript, I suggest to add these potential mechanisms to the model in Fig. 6 and to the discussion. Perhaps also of interest may be the earlier observation that host cell organelles can be found in the lumen of SIF, as reported by Krieger et al. (doi:10.1371/journal.ppat.1004374). The entrapment of vesicles and organelles required formation of double membrane SIF, a phenotype also dependent on function of SseF.

Reviewer #3

(Remarks to the Author)

The authors greatly improved the article with the changes made in the manuscript and figures, especially with the inclusion of more details regarding the methods employed and with data regarding mitochondrial ROS and host metabolism rewiring linked to anti microbial response. They have addressed most of my initial concerns.

Reviewer #4

(Remarks to the Author)

REVIEWER COMMENTS

Reviewer #1

In this study, the authors showed that the mitochondrial tricarboxylate carrier protein (CIC) is recruited to the SCV in *Salmonella*-infected cells. They proposed CIC to play a role in limiting ROS production by the host, promoting *Salmonella* intracellular survival and growth. Finally, they identified the SseF SPI-2 effector to be involved in the recruitment of CIC to the SCV.

The results are very interesting and particularly important for understanding the intracellular mechanisms by which *Salmonella* adapts to its vacuolar environment. The proposed model is essentially based on microscopy, infection, regulation and localisation experiments aimed at demonstrating that the presence of CIC on the SCV significantly reduces ROS production by the host and limits its antibacterial activity against intracellular *Salmonella*.

There is no doubt about the quality of the experiments and their novelty. However, I have a few comments, most of which focus on Figure 3, which presents results that are central to arguing and validating the main message and title of this article. From my point of view, a certain confusion and uncertainty persists after reading this section.

1. The wording, the legends and the organisation of the figures are rather confusing and unintuitive. Overall, it would be good to describe the results more explicitly to help the reader follow the experimental progress. In addition, the names of the probes studied or the reporters used are specified in the legends but not on the figure, which complicates and slows down understanding of the figure. Finally, the panels are not always numbered in the right order.

We thank the reviewer for this valuable feedback regarding the clarity and organization of the figures. We have revised the manuscript and figure legends to describe the experimental results more explicitly and to improve the narrative flow. In addition, we have reorganized the figure panels to follow a more logical sequence and have corrected the panel numbering to ensure consistency throughout the manuscript. The figures themselves have also been updated to include clear labels for all probes and reporters. These changes were made to enhance readability and facilitate the reader's understanding of the experimental design and results.

2. ROS are measured using the CM-H₂DCFDA indicator, which does not allow the type(s) of ROS measured to be distinguished. It would therefore be appropriate to use other probes or biosensors specific to one form of ROS and to specify which one, even if a soxS reporter is used.

We thank the reviewer for the insightful comment regarding ROS detection. In the revised manuscript, we addressed this concern by incorporating additional analyses to better distinguish the types of ROS encountered by vacuolar *Salmonella*. While CM-H₂DCFDA was initially used to detect overall oxidative stress in infected host cells, we complemented this approach with transcriptional reporters specific to distinct ROS species sensed by intracellular *Salmonella*. In the original manuscript, we used a *soxS* reporter to detect superoxide, which is one of the first ROS generated in the phagosome and is derived from NADPH oxidase activity. To assess hydrogen peroxide levels, we introduced biosensors driven by the promoters of the hydrogen peroxide-responsive transcriptional regulator *oxyR* and one of its downstream targets, *katG*. As shown in the revised Figures 3h-j, both *soxS* and *oxyR/katG* reporters exhibited increased activity in CIC-deficient cells, indicating elevated levels of superoxide and hydrogen peroxide within the SCV. Furthermore, to evaluate hydroxyl radical production, we tested the effects of iron chelators (DFO and polyP), which block Fenton chemistry. Treatment with these chelators significantly reduced ROS levels and rescued intracellular *Salmonella* replication in CIC-deficient cells (Figure 4h and 4i), consistent with the

presence of hydroxyl radicals generated via Fenton reaction. Taken together, these results support the conclusion that CIC deficiency leads to elevated levels of superoxide, hydrogen peroxide, and presumably hydroxyl radicals within the SCV.

3. The qRT-PCR measurements carried out on the katG gene show a relatively modest increase (about 1.2-fold 6h pi and 2.8-fold 24h pi) which is not in agreement with the differences in ROS levels observed in figures 3b (about 20-fold) and 3d (about 8-fold). Do the authors have an explanation?

We thank the reviewer for this thoughtful observation. Indeed, *katG* mRNA measured by RT-qPCR showed only modest increases, rising to about 1.2-fold at 6 h and 2.8-fold at 24 h post-infection, whereas ROS levels rose much more sharply, as shown with CM-H₂DCFDA (new Fig. 3d,e). In the revised manuscript, we also monitored *katG* expression using a *katG*-promoter reporter and obtained similar results (new Fig. 3j). Several factors may account for this apparent discrepancy:

1. Distinct ROS response pathways: KatG is primarily regulated by OxyR in response to hydrogen peroxide, whereas CM-H₂DCFDA and the *soxS* reporter reflect broader or different ROS species, including superoxide and hydroxyl radicals. Therefore, *katG* expression may not linearly reflect total ROS or superoxide levels in this context.

2. Spatial regulation: ROS production within infected cells is likely to be tightly compartmentalized. In our study, we observed a significant increase in total cellular ROS in CIC-deficient cells only following *Salmonella* infection, indicating that infection alters the host cell redox state. However, the biosensors and reporter systems used in this study were designed to specifically monitor the vacuolar environment harboring intracellular *Salmonella*. Consequently, ROS levels detected within the SCV may diverge from the total cellular ROS measured by CM-H₂DCFDA staining, which could account for the differences in magnitude observed. Moreover, the transcriptional response of *katG* likely reflects only one component of the broader oxidative stress adaptation. Additional detoxifying enzymes and regulatory pathways may be activated but not captured by *katG* expression alone, which may further explain the difference between ROS measurements and the modest fold-change in *katG* mRNA levels.

However, in the revised manuscript, we developed NanoLuciferase-based biosensor systems driven by the *oxyR* and *katG* promoters (shown in the new Fig. 3i and 3j), which more accurately reflect the oxidative stress response of vacuolar *Salmonella* compared to conventional transcriptional analyses.

4. DPI is an inhibitor of flavoenzymes, including NADPH oxidase. But it would probably be more convincing to carry out these same experiments in gp91^{-/-} macrophages lacking the catalytic subunit of this complex and therefore unable to produce superoxide anion or any other ROS.

We thank the reviewer for this helpful suggestion. While we did not use gp91-deficient macrophages, we addressed this point by employing a gp91^{phox}-specific inhibitory peptide (gp91ds-tat), which selectively inhibits NADPH oxidase by interfering with its assembly. As shown in the revised Fig. 3g, treatment with gp91ds-tat significantly restored intracellular *Salmonella* growth in CIC-deficient macrophages. This finding further supports a key role for NADPH oxidase in mediating ROS production that restricts bacterial replication within the SCV.

Minor points

Lines 113-115: This result is not supported by Figure S3 where the addition of CIC was not significant in the liver and spleen.

We thank the reviewer for pointing this out. In response, we expanded the number of mice used in the experiment to increase statistical power. As shown in new Supplementary Fig. 6, treatment with the CIC inhibitor resulted in a statistically significant reduction of *Salmonella* burden in the spleen. Although the reduction observed in the liver did not reach statistical significance, there was a clear trend toward decreased bacterial load. These results suggest

that while CIC inhibition has a measurable impact on systemic *Salmonella* infection, further optimization, such as increasing the inhibitor concentration or adjusting the dosing frequency, may be necessary to achieve more consistent therapeutic effects across different organs.

Lines 192-193: This is highly speculative and should be mitigated: all the experiments used to support this conclusion are indirect.

We agree that the evidence supporting the involvement of iron consumption via the Fenton reaction within the SCV is based on indirect measurements. In response, we have revised the wording of our conclusions in the manuscript to more accurately reflect the limitations of the current data.

Reviewer #2

Comments to the authors:

The manuscript Fu et al. proposes a novel mechanism of recruitment of host cell mitochondrial citrate transporter CIC to the *Salmonella*-containing vacuole (SCV) in *Salmonella*-infected mammalian cells. The work shows association of CIC with SCV membrane, and the authors propose functional insertion of CIC in the SCV membrane to reduce citrate concentrations in the SCV lumen. Reduction of citrate in the SCV in turn should limit Fenton chemistry reactions and protect intracellular *Salmonella* from oxidative damages. SPI-2 effector protein SseF is proposed to mediate the recruitment of CIC for inner mitochondrial membrane to the SCV membrane. This is an interested novel observation, but a large number of experimental limitations and biased interpretations limits my enthusiasm about this work.

Specific comments:

In Fig. 1, no description of the magnetic conjugation and separation protocol is provided. According to the legend, Salmonella were conjugated with magnetic beads, then used in invasion of HEK293T cells, followed by magnetic isolation of Salmonella containing compartments. Although the description of materials and methods is rather comprehensive for other parts of the manuscript, this procedure is not described at all. It appears technically rather challenging to conjugate magnetic beads to Salmonella in a way that is compatible to cell invasion. Please provide sufficient details for the procedure and show controls.

We thank the reviewer for highlighting this important point. The magnetic labeling and SCV isolation protocol used in our study was adapted from a previously published method (PMID: 30079117) with minor modifications optimized for our experimental system. In the revised manuscript, we have included a detailed description of this protocol in the Materials and Methods section under a newly added subsection titled "Isolation of *Salmonella*-containing vacuoles." This section describes the protocols for magnetically labeling *Salmonella*, conducting invasion assays in HEK293T cells, and isolating *Salmonella*-containing vacuoles (SCVs) via magnetic enrichment. To ensure that magnetic labeling does not interfere with the intracellular behavior of *Salmonella*, we conducted control experiments to assess intracellular replication. The result is presented in revised Supplementary Fig. 1a, and demonstrates that magnetic labeling does not impair bacterial intracellular replication. Furthermore, to validate that the labeled *Salmonella* maintain functional SPI-2 activation, we employed a reporter strain expressing superfolder GFP (sfGFP) under the control of the *pagC* promoter. *pagC* is regulated by the PhoP–PhoQ two-component system, which is activated in vacuolar conditions characteristic of the SCV. We observed comparable *pagC*-driven sfGFP expression in both non-labeled and magnetically labeled *Salmonella* (new Supplementary Fig. 1b), supporting the conclusion that magnetic labeling does not alter the intracellular localization or activation status of SPI-2-dependent gene expression. Together, these results demonstrate that magnetic labeling is compatible with *Salmonella* intracellular replication and proper vacuolar localization, validating the use of this method for studying the SCV during infection.

The visualization and quantification of CIC association with SCV (Fig. 1cde) requires controls. The association of LAMP1-positive compartments (late endosomes/lysosomes) with CIC-positive compartments needs to be determined under comparable conditions for non-infected cells and cells infected with SPI-2-deficient Salmonella.

We thank the reviewer for this insightful suggestion. In response, we have performed additional experiments to clarify the specificity of CIC recruitment to the SCV. As shown in the revised Fig. 1d and 1e, we observed a significant increase in CIC association with SCVs in cells infected with wild-type (WT) *Salmonella*, but not with the SPI-2-deficient $\Delta spiA$ mutant. To further evaluate the specificity of CIC localization, we examined its association with LAMP1-positive compartments under mock conditions, as well as in cells infected with WT or $\Delta spiA$ *Salmonella*. Quantitative analysis showed that CIC signal overlapping with LAMP1-positive endosomes and lysosomes remained low and comparable across all three conditions (new Fig. 1f). These results indicate that CIC recruitment is specifically induced by *Salmonella* infection in a SPI-2 type III secretion system-dependent manner and is not due to general colocalization with late endocytic or lysosomal compartments.

Does the protocol used for expansion microscopy also allow expansion of the bacterial cells? In Fig. 1 the labeled host cell compartments appear expanded, but not the bacteria within the SCV. If this is the case, the interpretation of potential contact between SCV membrane positive for LAMP1 and mitochondrial membranes positive for CIC has to be taken with care. To make valid conclusions about a potential colocalization of CIC with CIC membranes, or even the integration of CIC into SCV membranes, more detailed analyses are required.

We thank the reviewer for this insightful comment. Under current expansion microscopy (ExM) protocols, bacteria expand far less than host membranes (PMIDs 36097171, 38049404). Because the SCV membrane is host-derived and integrated into the endolysosomal network, we assessed its structural fidelity after expansion. Distortion analysis (Supplementary Fig. 2a–d) and root-mean-square error measurements ($< 0.1 \mu\text{m}$ over $5 \mu\text{m}$) reveal minimal deformation of host membranes, supporting the accuracy of CIC localisation on LAMP1-positive SCVs. Independent biochemical data corroborate the microscopy: magnetically isolated SCVs are enriched for CIC alongside LAMP1 (Fig. 1a,b), confirming that CIC is physically associated with the vacuolar membrane. We nevertheless recognise that differential expansion limits the spatial resolution of ExM and have added an explicit caveat in the Discussion. Together, these controls and orthogonal assays demonstrate that bacterial under-expansion does not compromise the conclusion that CIC localises to LAMP1-positive SCV membranes.

A Salmonella strain set was used that suffers from atypical cell division (septation defect). Note the appearance of filamentous Salmonella cells in Fig. 1 and Fig. S9. This phenomenon has been observed previously and was explained by the His auxotrophy of SL1344. Specific care should be taken in the interpretation of CFU assays for intracellular survival and replication, as filament formation will result in incorrect CFU counts.

We thank the reviewer for bringing up this important point. We acknowledge that *Salmonella* SL1344 carries a known His auxotrophy, which has been associated with filamentation under certain conditions. To address this concern, we also utilized the *Salmonella* 14028s strain in the experiments presented in the revised Fig. 2d. This strain does not display the same auxotrophy or filamentation phenotype under our experimental conditions (PMID: 15773994). The results obtained using 14028s were consistent with those from SL1344, reinforcing the robustness of our conclusions. Furthermore, we employed a fluorescence-dilution reporter (pFCcGi) to assess intracellular replication (Fig. 2b and 2c), and the results from this independent assay aligned with our CFU-based data, supporting the validity of our replication measurements.

Fig. 2: What is the effect of CIC k/o and CIC inhibitor on physiology of host cells? Give that the mitochondrial transporters are central for mammalian cell physiological, critical

assessment is required. Is host cell viability altered? Is phagocytic uptake or invasion by Salmonella affected? How is fatty acid metabolism altered in k/o cells or cell exposed to the inhibitor?

We thank the reviewer for this important comment regarding the physiological impact of CIC knockout and CIC inhibition on host cells. In the revised manuscript, we conducted additional experiments to provide a more comprehensive assessment of host cell physiology. We assessed the viability of CIC KO macrophages and CTPi (CIC inhibitor)-treated cells under both resting and infected conditions using a Zombie Green Fixable Viability Kit. No significant differences in viability were observed compared to parental controls (new Supplementary Fig. 3a). To evaluate whether CIC loss affects bacterial entry, we quantified intracellular *Salmonella* at 1 hour post-infection (the entry phase). Results showed no significant differences between CIC-deficient and control cells, indicating that neither CIC KO nor CTPi treatment impairs phagocytic uptake or bacterial invasion (new Supplementary Fig. 3b). These findings suggest that CIC loss does not compromise host cell viability or phagocytic function.

Given CIC's established role in citrate export and lipogenesis, we further investigated fatty acid metabolism under infection conditions. RT-qPCR analysis of ACLY (ATP citrate lyase) and BODIPY staining of lipid content revealed no significant differences between CIC-deficient or CTPi-treated cells and controls (new Supplementary Fig. 4b and 4c). Moreover, CRISPR/Cas9-mediated knockout of ACLY did not affect intracellular *Salmonella* replication (new Supplementary Fig. 4d-g), indicating that reduced lipogenesis is not responsible for the observed phenotype.

Together, these results support the conclusion that the defect in intracellular *Salmonella* replication is due to CIC's specific role in modulating the vacuolar environment, rather than general host cell dysfunction.

Fig. 4 fg: pHrodo-labelled Salmonella were used to monitor the SCV pH in dependency of CIC function. This is basically a useful experimental approach. Criticism here are in particular: a) if bacteria are prelabelled with a pH sensor, intracellular replication of Salmonella will gradually dilute the label. This issue may be limited at 6 h after infection (p.i.), but more prominent at 24 h (experiments shown in Fig. 1 demonstrate the dilution during replication in RAW macrophages). Thus, in absence of normalization, the pHrodo cannot be interpreted. b) As I understand from the description of this experiment, no normalization for loading of RAW cells with Salmonella was performed. Again, interpretation of the signal intensities is hardly possible. If the aim here is to correlate the CIC function to phagosomal pH, I suggest to use pHrodo labelled beads (that does not replicate within host cells).

We thank the reviewer for this thoughtful and constructive critique regarding the use of pHrodo-labeled *Salmonella* to monitor SCV pH. We agree that using pHrodo-labeled inert particles, such as beads, represents a robust strategy to assess phagosomal acidification independent of bacterial replication. We revised our experimental approach to avoid the potential confounding effects of bacterial replication on pHrodo signal. Specifically, we employed pHrodo Green AM, a membrane-permeable pH-sensitive dye that becomes increasingly fluorescent as pH decreases. We treated both mock- and *Salmonella*-infected cells with this dye and quantified fluorescence intensity within lysosomes and SCVs, which were identified using LysoTracker and *Salmonella* harboring a plasmid expressing mCherry, respectively. As shown in the revised Fig. 4d and 4e, we observed a significant increase in SCV acidity in CIC-deficient cells compared to the parental macrophage cell line, while lysosomal acidity remained unchanged between the two conditions. This suggests that the observed acidification is specific to the SCV rather than a general defect in endolysosomal acidification. To further support this observation, we utilized a Nanoluciferase-based biosensor under the control of the *stm1485* promoter, which responds to increased acidity within the SCV. As shown in the revised Fig. 4c, expression of the *stm1485*-driven reporter was markedly elevated in CIC KO cells compared to parental controls. Together, these data indicate that CIC deficiency specifically increases SCV acidity without affecting lysosomal pH, supporting a role for CIC in regulating vacuolar pH during infection.

X and Y scaling are missing in Fig. 4f (as for most flow cytometry histograms). We have added X and Y axis labels to all flow cytometry histograms to improve data presentation and readability.

*Fig. 4h-o: Here the authors analyse the expression of iron-regulated genes in intracellular Salmonella, and the data indicate iron limitation for Salmonella in CIC k/o cells. As other analyses were performed with Salmonella as biosensors, it would be of interest to use a biosensor approach also for analyses of iron limitation of Salmonella within the SCV. In the revised manuscript, we have incorporated a biosensor-based approach to directly assess iron limitation in intracellular Salmonella. Specifically, we utilized Salmonella strains expressing Nanoluciferase under the control of either the *iroB* or *tonB* promoter, which are involved in siderophore biosynthesis and siderophore-iron complex uptake, respectively. As presented in the new Fig. 4f and 4g, the reporter activity indicated significantly increased iron limitation in vacuolar Salmonella residing in CIC KO cells compared to parental control cells.*

Fig. 4: Is there any evidence for increased Fenton chemistry in CIC k/o cells? If this assumption is valid, oxidative damages should be more frequent in CIC k/o cells independent from Salmonella infection.

No study has reported elevated Fenton chemistry in CIC-deficient cells, and our data confirm this. In uninfected CIC-KO macrophages, total cellular ROS remained at baseline (revised Fig. 3d,e), showing that oxidative stress is not an intrinsic feature of CIC loss. The picture changes during infection: CIC-KO cells harboring *Salmonella* exhibited a pronounced, iron-dependent rise in vacuolar ROS (revised Fig. 4f,g). Chelating labile iron with deferoxamine or medium-chain polyP significantly restored bacterial viability (Fig. 4h,i), suggesting that excess iron drives Fenton reactions inside the SCV. Furthermore, biosensor measurements revealed higher citrate and greater acidification in CIC-deficient SCVs (revised Fig. 4b,c). We therefore propose that CIC normally exports citrate from the SCV into the cytosol; without this efflux, citrate accumulates, the lumen acidifies, and iron remains soluble, collectively fueling Fenton chemistry. This model explains why CIC-KO cells show no oxidative stress under basal conditions yet experience infection-specific ROS surges. Thus, CIC's protective function is context-dependent, becoming critical only in the presence of intracellular *Salmonella*.

The model also proposes that citrate export from the SCV by CIC is critical for the protection of Salmonella against oxidative damage. To appreciate the important of citrate concentration in the SCV, it would be important to determine the citrate concentration in the SCV in WT and CIC k/o cells, as in response to intracellular activities of Salmonella. Without such direct data, the models on the role of citrate in the SCV are rather speculative.

We thank the reviewer for highlighting this important point and fully agree that direct quantification of citrate within the SCV would substantially strengthen our conclusions. However, due to current technical limitations, accurately measuring metabolite concentrations within SCV compartments remains challenging. To address this, we employed an indirect approach using a transcriptional biosensor driven by the *citS* promoter. *citS* encodes a citrate transporter whose expression is upregulated in response to extracellular citrate availability, making it a functional indicator of environmental citrate levels. In our study, the *citS* reporter exhibited significantly elevated activity in CIC-deficient cells compared to parental controls (in new Fig. 4b), supporting the notion of increased citrate accumulation within the SCV. While this method does not provide absolute citrate concentrations, it offers biologically relevant evidence that *Salmonella* perceives higher citrate levels in the absence of CIC.

Fig. 5: The data shown in Fig.5 should demonstrate that SPI-2 effector SseF is mediating CIC recruitment to the SCV. This assumption of mainly based on a previous BiOLD approach, but not by systematic analyses of contribution of the various SPI-2 effectors.

We thank the reviewer for this insightful comment. While our hypothesis that SseF mediates CIC recruitment to the SCV was initially informed by a previously published BioID study identifying CIC as a putative SseF interactor (PMID: 31611645), we have conducted additional experiments to directly assess the specificity and functional contribution of SseF in our system. As shown in Fig. 5a (using a stable cell line constitutively expressing CIC-Myc) and Fig. 5b (using a knock-in cell line expressing endogenous CIC fused to a Myc tag), co-immunoprecipitation assays confirmed a specific interaction between CIC and SseF in infected cells. To further assess the specificity of this interaction, we examined two additional SPI-2 effectors, SopD2 and PipB2. However, no interaction between CIC and either of these effectors was observed under the same experimental conditions (new Supplementary Fig. 10a and 10b). Moreover, deletion of another core SPI-2 effector, SseG, did not disrupt the formation of the CIC-RAB7 complex (Supplementary Fig. 11d), further supporting a distinct and specific role for SseF. While a systematic analysis of all SPI-2 effectors is beyond the scope of this study, our combined biochemical and functional data provide strong support for the conclusion that SseF plays a central role in mediating CIC recruitment to the SCV. We have clarified this point in the revised manuscript.

Complementation of the phenotype of the sseF mutant strain shown in Fig. 5b is required.

We performed a complementation assay by expressing HA-tagged SseF in the Δ sseF mutant strain, as shown in the new Fig. 5c, d. The results demonstrate that complementation with SseF fully restores CIC recruitment to the SCV, supporting the role of SseF in mediating this interaction.

Experiment shown in Fig. S8 should demonstrate that SPI-2 effector proteins SopD2 and PipB2 are not involved in CIC recruitment to SCV. However, the IP results are difficult to compare to IP data shown in Fig. 5. For SseF, the HA tag was used for IP, and Fig. 5 indicates that SseF-HA is efficiently translocated into host cells and recovered by IP. This does not apply to SopD2-FLAG and PipB2-FLAG, as Figure S8 indicated very low levels of translocated effects. The experiment needs to be repeat with HA-tagged effectors to allow comparison and proper interpretation.

We thank the reviewer for this helpful suggestion. To address the concern, we repeated the co-immunoprecipitation assays using HA-tagged versions of SopD2 and PipB2, ensuring consistent tagging with the SseF-HA used in Fig. 5a. These constructs were expressed in the respective *Salmonella* mutant strains, and the results are shown in the new Supplementary Fig. 10a and 10b. Our findings demonstrate that neither SopD2-HA nor PipB2-HA interacts with CIC, further supporting the specificity of the CIC-SseF interaction.

All conclusions on CIC localization are based on transiently or permanently transfected host cells expressing tagged allele of CIC. It will be important to corroborate the findings made in the overexpression system with analyses of CIC expressed at endogenous levels, either using antibodies against CIC or an endogenously tagged allele of CIC.

We thank the reviewer for this valuable suggestion. Due to the lack of a suitable antibody for detecting CIC by immunostaining, we generated a knock-in cell line using CRISPR-Cas9 technology in which CIC is endogenously tagged with a Myc epitope. Using expansion microscopy (ExM), we observed clear recruitment of endogenously tagged CIC to the SCV, as shown in the new Supplementary Fig. 2e-h. Furthermore, we performed co-immunoprecipitation assays in this knock-in cell line and confirmed the interaction between endogenous CIC and SseF during infection (new Fig. 5b). Taken together, these results corroborate our findings from overexpression systems and provide strong evidence that CIC is recruited to the SCV at endogenous expression levels.

Fig. 6 and overall model: CIC is located in the inner mitochondrial membrane, and the model developed by the authors propose that CIC is recruited to the SCV membrane. Such mechanism would require extraction of CIC from the mitochondrial inner membrane, transfer

to the SCV membrane, and membrane insertion. The scheme in Fig. 6 proposes formation of vesicles that also contain Rab7 and SseF. How can such compartment be formed?

We appreciate the reviewer's insightful question. Our data clearly show that CIC accumulates on the SCV in an SseF-dependent manner, yet the precise trafficking route and molecular machinery remain to be defined. Several emerging pathways could plausibly deliver CIC from mitochondria to the vacuole. One possibility is that mitochondrial-derived vesicles (MDVs). MDVs selectively package inner-membrane cargo and frequently require RAB7 for their fusion with late endosomes/lysosomes. Under stress conditions, such vesicles could ferry CIC to RAB7-positive SCVs. Recent studies also describe mitochondrial-derived compartments (MDV) that shuttle specific membrane proteins to endolysosomal sites for turnover. A similar route, co-opted by SseF in coordination with RAB7, might mediate CIC delivery. Another possibility comes from recent reports of mitochondrial remodeling that generates structures positive for the outer mitochondrial membrane (SPOTs) and spanning OMM-IMM bridges (PMID: 35025629). Vesicles budding from these sites could release mitochondrial cargo, including CIC, into the cytosol for subsequent capture by the SCV. Taken together, our findings place SseF at this interface, suggesting that it promotes or stabilizes RAB7-positive contact sites or vesicular intermediates that translocate CIC from mitochondria to the SCV. In future studies, combining live-cell imaging of MDV markers with real-time tracking of mitochondrial membranes will help elucidate this trafficking pathway during infection.

Reviewer #3

In the article entitled 'Intracellular Salmonella hijacks the mitochondrial citrate carrier to evade host oxidative defenses', Fu et al present a compelling study investigating the role of the mitochondrial citrate carrier (CIC) in the intracellular survival and replication of Salmonella Typhimurium. The authors demonstrate that CIC is recruited to the Salmonella-containing vacuoles (SCVs), where it plays a crucial role in mitigating oxidative stress. This recruitment is mediated by the Salmonella effector protein SseF and the host GTPase Rab7. The authors state CIC alleviates oxidative damage by modulating citrate levels within the SCVs, thereby promoting bacterial survival and replication. Pharmacological inhibition of CIC significantly reduces bacterial replication and colonisation, suggesting CIC as a potential therapeutic target.

While the study provides valuable insights into host-pathogen interaction eventually introducing novel host-directed strategies for Salmonella, my enthusiasm for the manuscript would greatly increase if several critical shortcomings were addressed. For example, alternative hypotheses to CIC deletion impact on host metabolism and related antimicrobial response are not explored. The data generally support the claims, but additional evidence is needed for certain mechanistic aspects (e.g., citrate levels, SseF domains).

Major:

1. A main concern is that the description of some experiments lacks sufficient detail, making it challenging for the reader to follow. For example, the explanation of the fluorescence dilution experiments in Figure 2 is overly vague. There is no clarification of the two colours used, which one is inducible, what the fluorescent dilution signifies, or what the graphs show. Additionally, it is somewhat unclear how fold replication was calculated in this figure and whether this method is validated when using this tool. The figure 1 microscopy images don't look compelling, perhaps a better description would help.

We thank the reviewer for pointing out the need for clearer methodological descriptions and figure explanations. In the revised manuscript, we have updated the relevant sections of the Results and Materials and Methods to provide more detailed information on the fluorescence dilution assay shown in new Figure 2. The pFCcGi dual fluorescence plasmid used in this assay has been widely applied to study intracellular *Salmonella* replication and survival (PMID: 23592259; 31862381; 39374386). In this system, mCherry is constitutively expressed, while GFP is inducible by arabinose. After the removal of arabinose, actively replicating bacteria

progressively dilute the GFP signal, whereas non-replicating bacteria retain both mCherry and GFP signals. Therefore, the extent of GFP dilution serves as an indicator of bacterial replication at the single-cell level. We have also revised the figure legend to clearly explain the significance of fluorescent signal used for flow cytometry analysis. Additionally, we clarified that bacterial fold replication was calculated by comparing CFU at 24 hours post-infection to CFU at 1 hour post-infection, consistent with established protocols (PMID: 19364837; 27815887). Finally, we improved the figure legend and added annotations in Figure 1 to enhance the clarity and visualization of CIC recruitment to SCVs. We believe these revisions will help readers better follow the experimental design and interpretation of our findings.

2. Although the authors link citrate regulation to ROS mitigation, there is no direct measurement of citrate levels in SCVs or their correlation with ROS production. Similarly, while iron depletion is suggested as a key factor, the authors provide only indirect evidence. The authors should isolate SCVs and employ metabolomic or targeted biochemical assays to measure citrate and iron concentrations. Additionally, to confirm the role of the Fenton reaction, direct inhibition using polyphosphate (a potent inhibitor in vitro) should be tested (10.1128/mbio.01017-20).

We thank the reviewer for this valuable suggestion. To strengthen our conclusion that citrate levels are elevated within the SCV in CIC-deficient cells during infection, we employed a transcriptional biosensor system driven by the *citS* promoter. The *citS* gene encodes a citrate transporter whose expression is known to be upregulated in response to increased environmental citrate availability, making it a sensitive and functional indicator of local citrate concentration. In our study, the *citS*-NanoLuc reporter exhibited significantly higher activity in CIC-deficient cells compared to parental controls (new Fig. 4b), supporting the notion of citrate accumulation within the SCV due to impaired citrate efflux in the absence of CIC.

Using a similar biosensor-based approach, we examined the expression of *iroB* and *tonB*, genes involved in siderophore biosynthesis and siderophore-iron complex uptake, respectively. These reporters, also driving NanoLuc expression, revealed increased activity in CIC-deficient cells (new Fig. 4f and 4g), consistent with enhanced iron limitation experienced by *Salmonella* in the SCV. These transcriptional changes serve as indirect but functionally relevant indicators of altered nutrient availability in the vacuolar microenvironment. To further test the hypothesis that Fenton reaction-driven ROS production is contributing to antimicrobial stress within the SCV, we used polyphosphate (polyP), an iron chelator, as suggested by the reviewer. Treatment with polyP significantly reduced oxidative stress in intracellular *Salmonella* and partially restored bacterial replication in CIC-deficient cells (new Fig. 4i). These findings are consistent with the interpretation that iron-dependent ROS production via the Fenton reaction contributes to bacterial growth restriction in the absence of CIC.

Together, these additional results support our original conclusions and reinforce the role of CIC in modulating SCV chemistry to limit oxidative stress and promote *Salmonella* survival in the vacuolar niche.

3. How does CIC inhibition/depletion, used in several experiments, affect citrate concentration in mitochondria? Citrate is involved in TCA cycle, its accumulation in CIC KO cells could lead to enhanced oxidative phosphorylation. Cell metabolism pathways such as glycolysis and OXPHOS (and mtROS) have been repeatedly linked to interferon response mounting (10.1016/j.cell.2019.05.003, 10.1126/scisignal.aaf1933, 10.3390/v16091391) which is known to inhibit Salmonella replication. It is then possible to argue that the inhibition observed in CIC KO cells is in part linked to a such effect.

We thank the reviewer for this insightful comment. Although mitochondrial citrate levels were unchanged (Supplementary Fig. 4a), we explored whether loss of CIC activity rewires host metabolism in a way that restricts *Salmonella*. Using Seahorse real-time metabolic flux analysis, we compared mitochondrial respiration and glycolytic flux in mock- and *Salmonella*-infected cells treated with vehicle or the CIC inhibitor CTPi. As shown in new Supplementary Fig.5, there were no significant differences in mitochondrial oxidative phosphorylation (OXPHOS) or glycolytic capacity between these groups, indicating that CIC inhibition does not

markedly alter basal energy metabolism under these conditions. To evaluate the potential involvement of mitochondrial ROS (mtROS), we measured mtROS levels in both mock-infected and infected cells. CIC depletion did not lead to elevated mtROS in uninfected cells (new Fig. 3a and 3b), although an increase in mtROS was observed upon *Salmonella* infection in CIC-deficient cells. Importantly, treatment with a mitochondrial ROS scavenger (MitoQ) did not rescue bacterial replication (revised Fig. 3c), suggesting that mtROS is not the primary factor responsible for restricting *Salmonella* growth in CIC-deficient cells.

Additionally, we examined activation of the interferon/STAT1 pathway, which has been implicated in antimicrobial responses linked to metabolic reprogramming. We observed no significant STAT1 activation in CIC-deficient cells compared to parental controls (shown in new Supplementary Fig. 13), indicating that interferon signaling is unlikely to explain the impaired bacterial replication phenotype.

Together, these results suggest that the restriction of *Salmonella* observed upon loss of CIC function is not driven by global alterations in host metabolism, mitochondrial ROS, or interferon responses, but rather by localized changes within the SCV environment that impair bacterial survival.

In a similar manner, citrate and TCA cycle are involved in the production of itaconate, a potent antimicrobial metabolite. It is possible to argue that accumulation of citrate in the mitochondria of CIC KO cells could lead to a change in itaconate production (10.1016/j.celrep.2022.110391) therefore limiting Salmonella replication.

To evaluate whether enhanced itaconate production contributes to the observed bacterial restriction, we treated both parental and CIC-deficient cells with an IRG1 inhibitor, which targets the mitochondrial enzyme responsible for itaconate synthesis. As shown in the revised manuscript (Supplementary Fig. 12), CRISPR/Cas9-mediated knockout of IRG1 failed to restore intracellular *Salmonella* replication in cells treated with the CIC inhibitor. This finding suggests that increased itaconate production is unlikely to account for the restriction of *Salmonella* replication in the absence of CIC. Therefore, the impaired bacterial growth appears to be independent of itaconate-mediated antimicrobial activity.

An alternative (or complementary) mechanism to the one presented in this work is highly possible and should be explored (or at least discussed) given that the level of Salmonella growth did not reach the level of parental cells after inhibition of cell ROS in CIC KO cells.

We appreciate the reviewer's suggestion to consider additional mechanisms limiting *Salmonella* replication in CIC-deficient cells. In our revised manuscript, we clarify that inhibition of host ROS production fully restores bacterial growth to parental levels when NADPH oxidase activity is blocked. Specifically, treatment with either diphenyleneiodonium (DPI) or the gp91phox-blocking peptide (gp91 ds-tat) completely rescues intracellular bacterial growth in the CIC-knockout line, matching that seen in parental cells (revised Fig. 3f,g). These results indicate that excess ROS, rather than other antimicrobial pathways, is the principal constraint on bacterial replication in the absence of CIC.

4. The authors strongly assert that SseF is the sole mediator of CIC recruitment. However, data from Figure 5B and Supplementary Figure S9 show that CIC recruitment persists, albeit at reduced levels, in the absence of SseF, indicating additional mechanisms. Revise the assertion regarding SseF's exclusivity in CIC recruitment. The role of other SPI-2 effectors or host factors should be discussed.

We thank the reviewer for this important comment. In the original manuscript we analyzed approximately 30 SCVs when assessing CIC recruitment. In the revised version, we expanded the dataset, as shown in the new Figure 5d, and demonstrate that CIC recruitment to the SCV is markedly reduced in the Δ sseF strain. Complementation of the mutant with sseF fully restored CIC localization, confirming that SseF is a key mediator of this process. We acknowledge, however, that additional bacterial or host factors may contribute to the residual CIC signal observed in the absence of SseF. Accordingly, we have revised the Discussion to

consider possible roles for other SPI-2 effectors or host trafficking pathways that could facilitate partial CIC recruitment when SseF is absent.

Also, the mechanistic details by which SseF and Rab7 facilitate CIC recruitment remain elusive. Mutational analyses of SseF to identify critical domains necessary for interaction, could be done or at least discussed.

We thank the reviewer for this insightful comment. We agree that a deeper understanding of how SseF and RAB7 direct CIC to the SCV would substantially strengthen our study. In the revised manuscript we now discuss the value of mutational analyses to pinpoint the regions of SseF required for CIC recruitment. As shown in new Supplementary Fig. 10c, deletion mapping reveals that amino acids 195-205 in the C-terminal region of SseF are essential for CIC binding. This result underscores the feasibility of future structure-function studies to define the precise SseF motifs that mediate effector-host interactions.

5. One of my major concerns is the inconsistency in the methods and cells used. It raises concerns about reproducibility and interpretation of the data. At several time in their work the authors changed the MOI used (MOI 5 in Figure 2, MOI 20 in Figure 3, MOI 100 in Figure 4...) for infection as well as the cells used for their assays, without justifying it or discussing it. These changes have fundamental consequences since the MOI used could affect the response elicited by the host. Furthermore, the infection in epithelial cells (HeLa) and non-epithelial cells (RAW264.7 and BMDM) is fundamentally different. An explicit rationale for the use of different MOIs and cell lines in each experiment should be provided. Moreover, the authors should discuss how these differences might influence host responses and Salmonella replication. Also, ensure that all experiments specify the cell type used. This would make it easier to interpret the results, which are currently confused.

We appreciate the reviewer's comment and have addressed it throughout the revised manuscript. The Materials and Methods now provides a detailed justification for the MOIs selected in each assay, the Results section explains our choice of RAW264.7 macrophages and HeLa epithelial cells, and the Discussion considers how both cell lineage and inoculum size can shape host-pathogen interactions. These additions clarify our infection parameters and enhance the reproducibility and interpretability of the study. A brief summary of these changes is provided below.

MOI usage: We now provide a clear rationale for the MOI used in each experiment, as described in the Materials and Methods section. Replication assays and measurement of ROS levels were conducted at low MOIs (5-10) to avoid artifacts linked to excessive bacterial burdens. Flow-cytometry experiments used an intermediate MOI of 20, which yielded enough infected cells for reliable detection and sorting without compromising cell integrity. To isolate intracellular *Salmonella* RNA and perform high-resolution fluorescence imaging, we used an MOI of 100 because this higher inoculum increased the proportion of infected cells, ensured adequate bacterial RNA yield, and produced a strong signal-to-noise ratio for microscopy while maintaining acceptable host-cell viability.

Cell line used in the study: We have carefully reviewed all figures and legends to ensure that the specific cell type used in each experiment is now clearly indicated. RAW264.7 macrophages and bone marrow-derived macrophages (BMDMs) were primarily employed to model intracellular *Salmonella* replication. In addition, we evaluated the role of CIC in the intestinal epithelial cell line Henle-407 and used HeLa cells for imaging-based assays, including confocal and expansion microscopy. The consistent phenotypes observed across these different cell types suggest that CIC plays a critical and conserved role during *Salmonella* infection. Different cell lines were selected based on the experimental requirements of each assay, and this rationale has now been clarified in the revised manuscript.

Minor:

1. Figure 1C: The authors stated that they noted the close proximity between SCV and mitochondria, as well as CIC localization on mitochondria in non-infected cells. However, no

mitochondrial marker was used in these experiments and images with mitochondrial marker should be provided.

We have repeated the experiments using CIC and a mitochondrial marker, and the updated results are now presented in the new Fig. 1c.

2. *Figure 1E: Quantitative data are presented, but the recruitment dynamics could be more clearly visualized. Add an illustrative image or graph summarizing CIC recruitment over time.* We have now included a representative image series in the revised Fig. 1d and Supplementary Fig. 2i that illustrates CIC recruitment to the SCV over time. These additions enhance the clarity of our findings and provide a more comprehensive view of the temporal dynamics of CIC recruitment.

3. *Figure 2G: The reduction in bacterial replication may result from cell death rather than inhibition of Salmonella. An assessment of the cytotoxicity of the CIC inhibitor at the concentrations used in the study, should be provided.*

In the new Supplementary Fig. 3a, we assessed the cytotoxicity of the CIC inhibitor on host cells and found no significant difference in cell viability compared to controls.

4. *Line 112: “in vivo” should be in italics.*

Thank you for pointing this out. We have corrected this.

5. *Figure 3: ROS is highly labile, making static measurements potentially misleading. Is it possible to consider live imaging for ROS measurements? Live imaging could therefore permit to have a better understanding of ROS dynamics in the different conditions tested.*

We thank the reviewer for this thoughtful suggestion. Live-cell imaging would indeed offer valuable insight into the real-time dynamics of ROS during infection. In the present study, we addressed ROS production with a panel of nanoluciferase-based biosensors that report superoxide, hydrogen peroxide, and hydroxyl-radical stress in vacuolar *Salmonella* (revised Fig. 3h–j). Although we did not include live imaging here, we recognize its potential and plan to adopt this approach in future work to track ROS kinetics under a wider range of conditions.

6. *Line 205: “in vivo” should be replaced by “in vitro”.*

Thank you for pointing this out. We have corrected this.

REVIEWER COMMENTS

Reviewer #1 (Remarks to the Author):

I have read the new results and the changes made by the authors. My suggestions have all been taken into account and my concerns resolved after a lot of experimental work and rewriting of many paragraphs.

I would like to congratulate the authors on their work which, in my opinion, makes their publication much sounder and a much more flowing read.

We greatly appreciate the reviewer's recognition of our efforts, which has helped us strengthen the scientific soundness and readability of our work.

Reviewer #2 (Remarks to the Author):

Thank you for the careful revision and detailed response to my comments. The authors have devised and executed a large number of additional experiments in order address the comments made to the original version of the manuscript. The new data added to the manuscript significantly improve the overall quality of the study and the robustness of interpretation.

For this reviewer, two points previously made still require attention:

1) Expansion microscopy for colocalization analyses: A correctly responded by the authors, the degree of expansion of host cell organelles and intracellular *Salmonella* is different. The method is use to address, with increased resolution, proximity between host cell membrane compartments. LAMP1-positive compartment can be *Salmonella*-containing vesicles, or host cell compartments that are not modified by activity of intracellular *Salmonella*. Thus, conclusions based on colocalization between LAMP1 and mitochondrial marker remain of limited value. I suggest to adjust the interpretation of this experiment and to tone down the statements based to expansion microscopy data. Alternatively, if the authors wish to build on expansion microscopy, markers other than LAMP1 would be required, for example SPI-2 effector proteins such as SseF that specifically insert in SCV and SIF membranes. As effector proteins as markers will only work in infected cells, comparison has to be made between wild-type and mutant strains.

We thank the reviewer for this comment. We agree that LAMP1 alone is not sufficient to distinguish *Salmonella*-containing vacuoles (SCVs) from other host compartments. In our analyses (Figure 1d), SCVs were defined as LAMP1-positive vesicles containing *Salmonella* DNA (DAPI-labeled), ensuring that only bona fide SCVs were evaluated. We have clarified this in the figure legend to avoid misunderstanding. In addition, we have toned down the interpretation of CIC colocalization with the SCV to reflect the technical limitations of expansion microscopy.

2) Recruitment of CIC to SCV: in the initial set of comments, I asked how recruitment of CIC from inner mitochondrial membranes to the SCV membrane can be mechanistically explained. The authors provide a set of potential routes such s formation of mitochondria-derived vesicles as potential vehicle. Although currently speculative and beyond the range of investigations in this current manuscript, I suggest to add these potential mechanisms to the model in Fig. 6 and to the discussion. Perhaps also of interest may be the earlier observation that host cell organelles can be found in the lumen of SIF, as reported by Krieger et al. (doi:10.1371/journal.ppat.1004374). The entrapment of vesicles and organelles required formation of double membrane SIF, a phenotype also dependent on function of SseF.

We have updated the discussion and Figure 6 to incorporate potential mechanisms by which CIC may be recruited from mitochondria to the SCV membrane. We also cited relevant work

describing the entrapment of host organelles in SIFs. These additions provide a broader mechanistic context while acknowledging that such routes remain speculative.

Reviewer #3 (Remarks to the Author):

The authors greatly improved the article with the changes made in the manuscript and figures, especially with the inclusion of more details regarding the methods employed and with data regarding mitochondrial ROS and host metabolism rewiring linked to anti microbial response. They have addressed most of my initial concerns.

We appreciate the reviewer's positive feedback and are glad the revisions resolved the concerns.

Reviewer #4 (Remarks to the Author):

We thank the reviewer for the comments.